# A *Pseudomonas* T6SS effector recruits PQS-containing outer membrane vesicles for iron acquisition

Jinshui Lin[1,2,*], Weipeng Zhang[1,3,*], Juanli Cheng[1,2], Xu Yang[1], Kaixiang Zhu[1], Yao Wang[1], Gehong Wei[1], Pei-Yuan Qian[3], Zhao-Qing Luo[4] & Xihui Shen[1]

Iron sequestration by host proteins contributes to the defence against bacterial pathogens, which need iron for their metabolism and virulence. A *Pseudomonas aeruginosa* mutant lacking all three known iron acquisition systems retains the ability to grow in media containing iron chelators, suggesting the presence of additional pathways involved in iron uptake. Here we screen *P. aeruginosa* mutants defective in growth in iron-depleted media and find that gene *PA2374*, proximal to the type VI secretion system H3 (H3-T6SS), functions synergistically with known iron acquisition systems. PA2374 (which we have renamed TseF) appears to be secreted by H3-T6SS and is incorporated into outer membrane vesicles (OMVs) by directly interacting with the iron-binding *Pseudomonas* quinolone signal (PQS), a cell–cell signalling compound. TseF facilitates the delivery of OMV-associated iron to bacterial cells by engaging the Fe(III)-pyochelin receptor FptA and the porin OprF. Our results reveal links between type VI secretion, cell–cell signalling and classic siderophore receptors for iron acquisition in *P. aeruginosa*.

[1] State Key Laboratory of Crop Stress Biology for Arid Areas and College of Life Sciences, Northwest A&F University, Yangling, Shaanxi 712100, China. [2] Shaanxi Engineering & Technological Research Center for Conversation & Utilization of Regional Biological Resources, College of Life Sciences, Yan'an University, Yan'an, Shaanxi 716000, China. [3] Division of Life Science, Hong Kong University of Science and Technology, Clear Water Bay, Hong Kong. [4] Purdue Institute of Immunology, Inflammation and Infectious Diseases and the Department of Biological Sciences, Purdue University, 915 West State Street, West Lafayette, Indiana 47907, USA. * These authors contributed equally to this work. Correspondence and requests for materials should be addressed to X.S. (email: xihuishen@nwsuaf.edu.cn).

Metal ions such as iron, copper and manganese are essential for virtually every life process by functioning as components of enzymes, regulatory protein and protein complexes involved in metabolism and signalling. Because of its unique redox potential, the transitional metal iron serves important roles as cofactors in enzymes essential for many cellular processes[1,2]. In higher organisms, iron often is stored by the formation of high-affinity complexes with proteins. The sequestration of iron by host proteins constitutes an important branch of host immunity to bacterial infection because pathogens need iron for both metabolism and virulence[2–4]. This arm race drives bacterial pathogens to evolve sophisticated mechanisms to scavenge this essential nutrient from the environment, including their hosts. Commonly used strategies include: the synthesis and secretion of siderophores, molecules that bind iron with high affinity in extracellular milieu and re-enter cells via specific transporters[5,6]; the use of receptors on cell surface to recognize and import haeme and haemoprotein, which after being shuttled across the cell wall, are degraded by a haeme oxygenase to release iron[7]; expression of proteins that directly target host iron-binding proteins, such as transferrin and lactoferrin[8,9]; and interference with the function of host proteins involved in iron sequestration[10]. It is important to note that a bacterial pathogen often utilize combinations of these strategies to effectively compete for iron to maximize their proliferation under specific conditions. For example, the intracellular bacterial pathogen *Legionella pneumophila* produces at least one siderophore (legiobactin), which functions to compete for iron from the environment[11]. Interestingly, a phagosomal membrane-embedded transporter that is translocated by its Dot/Icm type IV secretion system is important for its intracellular life cycle, presumably by importing iron from the host cytosol[12]. This discovery represents an example in which iron acquisition is directly linked to effectors secreted by a specialized protein secretion system dedicated to bacterial virulence.

*Pseudomonas aeruginosa* is a ubiquitous bacterial pathogen that is highly adaptive to nutrient challenges; its extremely robust metabolic systems allow it to survive by using dicarbon compounds such as acetate as the sole carbon source[13]. In parallel with its highly efficient carbon-assimilation ability, *P. aeruginosa* effectively competes for iron by at least three independent mechanisms. First, the bacterium produces pyoverdine and pyochelin, two siderophores, which bind iron with different affinities prior to being transferred into bacterial cells via the TonB-dependent receptors[14,15]. Recently, a nicotianamine siderophore-mediated iron uptake system was identified to be essential for the growth of *P. aeruginosa* in airway mucus[16]. Second, it imports haeme molecules from haemoproteins from hosts[17,18]. Third, the bacterium reduces $Fe^{3+}$ to $Fe^{2+}$ by producing phenazine and then imports $Fe^{2+}$ by the Feo system[19]. Finally, in niches occupied by multiple bacterial species, *P. aeruginosa* is capable of importing iron bound by siderophores released by other bacteria using multiple receptors[17].

Specialized protein secretion systems are essential for the interactions between bacteria and their environments, particularly in the context of infection whereby secreted virulence factors are required for thwarting host defense or for the acquisition of important nutrients[20,21]. The contact-dependent type VI secretion system (T6SS) is one of the at least seven such systems that is widely distributed in Gram-negative bacteria. T6SSs mostly are recognized for their role in interbacterial species competition or in the modulation of host processes for successful infection by delivering effectors into bacterial or host cells[22–27]. Multiple distinct T6SSs can exist in a given bacterium, which raises the possibility that some of these transporters assume noncanonical functions in the life cycle of the bacterium. Indeed, the T6SS-4 from *Yersinia pseudotuberculosis* is involved in the resistance to oxidative stress by secreting a metal ion-binding protein that imports zinc to mitigate reactive oxygen species[28]. The genome of *P. aeruginosa* strain PAO1 encodes three separate T6SSs called H1, H2 and H3. H1-T6SS is known to target prokaryotic cells by delivering multiple bacteriolytic toxins into target cells, providing a competitive advantage to *P. aeruginosa* in polymicrobial communities[29]. In contrast, H2- and H3-T6SS target both prokaryotic and eukaryotic cells by using the PldA and PldB trans-kingdom effectors[23,24]. Moreover, both H2- and H3-T6SS contribute to the virulence of *P. aeruginosa* in animal and plant infection models[30,31]. The expression of *P. aeruginosa* T6SSs is differentially regulated by quorum sensing (QS). Whereas the expression of H1-T6SS is suppressed by both the homoserine lactone transcription factor LasR and the 4-hydroxy-2-alkylquinoline transcriptional regulator MvfR, the expression of H2- and H3-T6SS is positively regulated by MvfR and LasR[31]. In addition, PqsE, a key component of the MvfR regulon, is required for the expression of part of H3-T6SS but not H2-T6SS[31]. However, the function of H2- and H3-T6SS remains largely unknown.

Here we demonstrate that PA2374, apparently secreted by H3-T6SS, is involved in iron uptake by interacting with outer membrane vesicles (OMVs) and the *Pseudomonas* quinolone signal (PQS) system.

## Results

### A gene adjacent to H3-T3SS is important for iron acquisition.

Mutants of *P. aeruginosa* defective in the pyoverdin, pyochelin or the Feo system are still able to grow in iron-deficient media[32,33]. To examine whether these three systems are all the iron acquisition mechanisms by this bacterium, we created a mutant defective in the pyoverdin biosynthetic pathway ($\Delta pvdA$), the pyochelin synthetase ($\Delta pchE$) and the ferrous iron transport ($\Delta feoB$). As expected, this mutant, designated PAΔ3Fe, still can grow in a minimal medium (MM) containing $0.5\,\mu g\,ml^{-1}$ iron chelator ethylenediamine-$N,N'$-bis(2-hydroxyphenylacetic acid) (EDDHA) (Supplementary Fig. 1). The ability of strain PAΔ3Fe to grow in iron-depleted media suggests the existence of yet unidentified iron acquisition systems. To identify the genes potentially involved in iron acquisition in this strain background, we established a genetic screening in which mutants of PAΔ3Fe induced by the Tn5 transposon were individually tested for growth in MM containing EDDHA. From the first 800 candidates tested, a mutant displaying severe growth defect was identified (Supplementary Fig. 1). Further analysis revealed that in the mutant the transposon inactivated gene *PA2374*, which is adjacent to the gene cluster coding for H3-T6SS (Fig. 1a). The close proximity of *PA2374* to H3-T6SS suggests that it may code for a substrate of this transporter, and we thus tentatively designated it as *tseF* (Type VI secretion system effector for Fe uptake).

On the chromosome of *P. aeruginosa* strain PAO1, *tseF* is situated at the distal end of the H3-T6SS locus and is next to *vgrG3*, which codes for a core component of the H3-T6SS (Fig. 1a); *tseF* encodes a 178 amino acid protein without detectable similarity to proteins of known function. This protein is conserved among different strains of *Pseudomonas* and its homologues are present in diverse bacteria, and most of them co-localized with a VgrG homologue (Supplementary Fig. 2). The deduced protein is predicted to contain three membrane occupation and recognition nexus motifs (Fig. 1b), which in eukaryotic proteins are known to be involved in the targeting of protein to specific organelles, probably by interactions with signature phospholipids found in biological membranes[34,35].

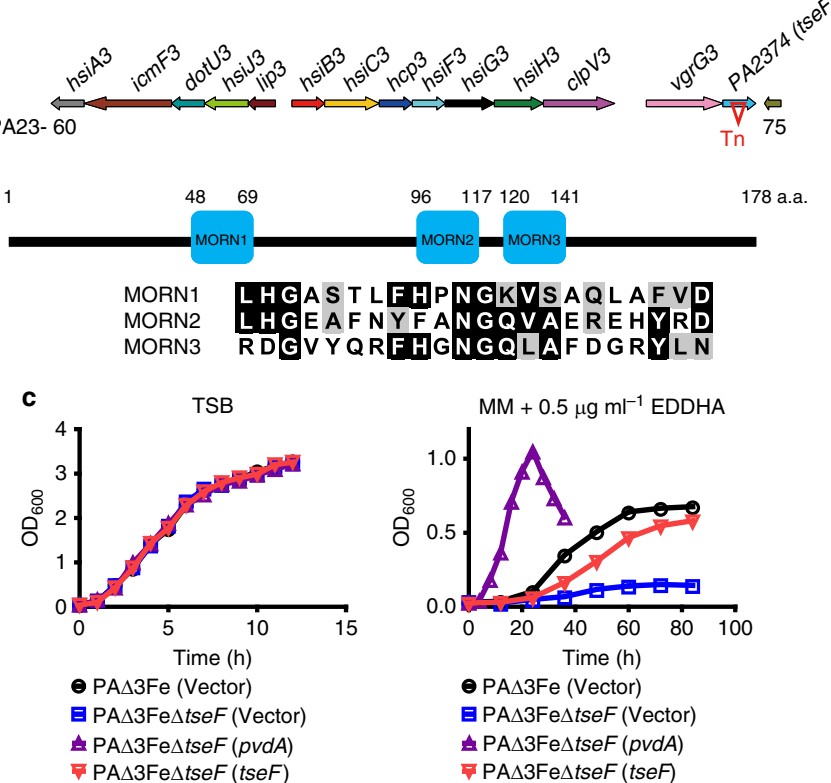

**Figure 1 | A gene proximal to H3-T6SS is required for the growth of a mutant missing known iron acquisition systems.** (**a**) The organization of H3-T6SS and its neighbouring genes. Structural genes are depicted as colour arrows that indicate the transcriptional directions. The gene numbers are only indicated at the end of the cluster. Gene *PA2374* (*tseF*) identified in the screening is marked by an inverted triangle, indicating transposon insertion. (**b**) Domain structure of TseF. The positions and regions of the predicted membrane occupation and recognition nexus motifs are shown. (**c**) The growth of the mutants in iron-deficient media. Relevant bacterial strains were grown in TSB or succinate MM containing EDDHA ($0.5 \mu g\, ml^{-1}$). Cell growth was monitored by measuring $OD_{600}$. Note that the deficiency of growth by mutant PAΔ3FeΔ*tseF* in iron-deficient media can be restored by expressing either *pvdA* or *tseF*. The curves represent three biological replicates; error bars are s.d.

To validate the phenotype associated with the original mutant, we constructed a mutant (PAΔ3FeΔ*tseF*) in which *PA2374* was deleted in the background of strain PAΔ3Fe. As expected, this strain failed to grow in iron-depleted media, similarly to the original transposon-induced mutant (Fig. 1c and Supplementary Fig. 1). Furthermore, the expression of *pvdA* or *tseF* restored the growth of PAΔ3FeΔ*tseF* in iron-limited media (Fig. 1c). Notably, the rescue of growth by *pvdA* is more robust than that by *tseF* (Fig. 1c), suggesting that, under our experimental condition, the PvdA system is more efficient in importing iron than the system participated by TseF.

**TseF is probably a substrate of H3-T6SS.** Genes coding for substrates of T6SSs often localize in regions proximal to structural genes, such as VgrG and Hcp[27,36]. To examine whether this protein is a substrate translocated by H3-T6SS, we expressed a TseF-VSVG fusion in a panel of bacterial strains with distinct genotypes. As a control, a VSVG fusion of the translocator Hcp is similarly constructed. TseF can be readily detected in culture supernatant of wild-type bacteria (Fig. 2a). Deletion of the essential components of the T6SS complex such as the ATPase *clpV3* or the outer sheath proteins *hsiB3-C3* drastically reduced but did not completely ablate the release of TseF into culture supernatant by the bacteria. The secretion of Hcp3 was similarly reduced but not abolished in these mutants (Fig. 2a). These results suggest that TseF may be a substrate of the H3-T6SS. To further substantiate this conclusion, we examined the interactions between TseF and VgrG, the T6SS component that acts as the

carrier for the secretion of effector by direct binding[27,37]. Glutathione S-transferase (GST)-TseF was able to retain VgrG3 and the VgrG homologue VgrG1b encoded by *PA0095* but not VgrG1a. In contrast, no interaction was detected between any of these VgrG proteins and PA0533, a protein not relevant to T6SS (Fig. 2b). Thus TseF is likely a substrate of the H3-T6SS; it selectively interacts with the VgrG carrier protein from some but not all T6SSs from *P. aeruginosa*. The somewhat promiscuous binding of TseF to more than one VgrG orthologues may account for its residual secretion by mutants lacking core components of the H3-T6SS. The secretion exhibited by H3-T6SS mutants suggests potential recognition of this protein by other secretion systems, possibly H1- and H2-T6SSs.

**PQS interacts with TseF.** The requirement of TseF for maximal bacterial growth in iron-depleted condition and its translocation by H3-T6SS suggest that this protein may participate in iron sequestration from the extracellular milieu, probably by directly binding to iron. Yet, we were unable to detect interactions between TseF and iron ions ($Fe^{3+}/Fe^{2+}$) (Supplementary Fig. 3), suggesting the use of a different mechanism for the involvement of this protein in iron uptake. We thus examined whether TseF interacted with molecules released by the bacterium that could be involved in iron uptake. To this end, we incubated GST-TseF with culture supernatants of *P. aeruginosa* grown in a MM and examined the potential effects on TseF by determining its mobility in native gels. We found that incubation with culture supernatants promoted the mobility of TseF (Fig. 3a).

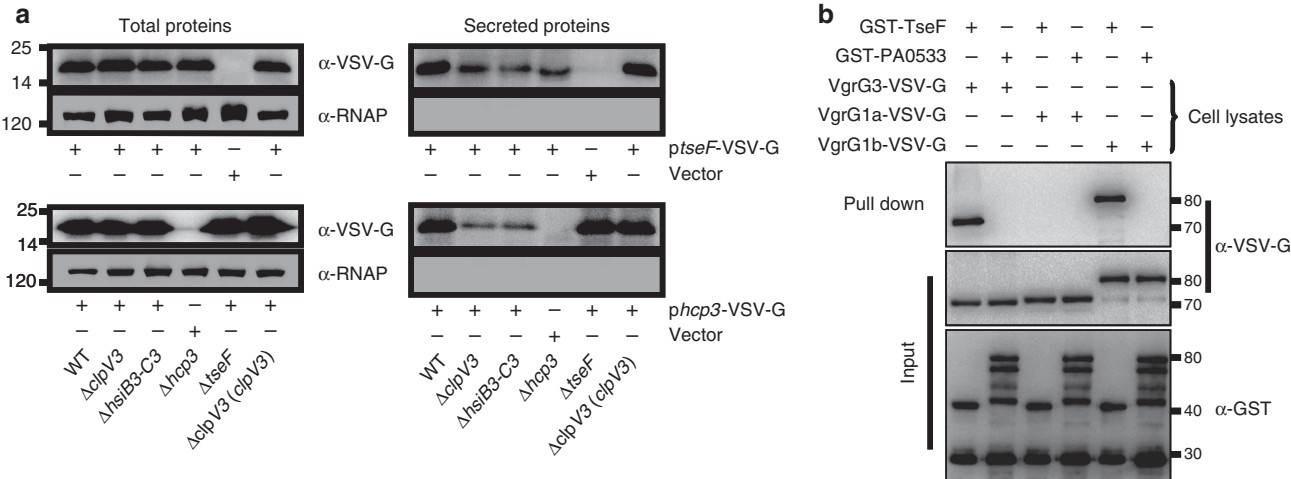

**Figure 2 | TseF is a substrate of H3-T6SS. (a)** A plasmid directing the expression of TseF-VSV-G chimera was introduced into the indicated *P. aeruginosa* strains. Total protein or proteins in culture supernatant was probed for the presence of the fusion protein. The cytosolic RNA polymerase (RNAP) was similarly detected as a control. Note that deletion of H3-T6SS component genes *clpV3*, *hsiB3-C3* or *hcp3* drastically reduced the release of TseF-VSV-G into extracellular milieu. **(b)** TseF interacts with VgrG3 and VgrG1b. GST-TseF was incubated with three different VgrG proteins, and protein complexes were captured by glutathione beads. Note that TseF can be co-purified with VgrG3 or VgrG1b but not VgrG1a. The unrelated protein PA0533 cannot be co-purified with any of the tested proteins. Full blots are shown in Supplementary Fig. 14.

Such increase did not occur when culture supernatants of *Escherichia coli* were used (Fig. 3a).

To identify the factors in culture supernatant responsible for promoting the mobility of TseF, we employed liquid chromatography mass spectrometry (LC-MS) to determine the compounds in supernatant extracts whose abundance (indicated by changes in their chromatograms) was reduced after incubation with GST-TseF. These experiments revealed that the abundance of a major peak corresponding to PQS (2-heptyl-3-hydroxy-4-quinolone; $m/z = 260$) was drastically reduced in supernatants that had been incubated with GST-TseF (Supplementary Fig. 4), suggesting that TseF titrates this compound from the culture supernatant, likely by binding. To validate this observation, we deleted the genes for the production of the PQS signal (*pqsA* and *pqsH*) and found that the supernatant of the mutant no longer caused alternations in the mobility of TseF (Fig. 3b). Consistently, deletion of the QS systems *las* and *rhl* abolished the ability of the culture supernatant to alter TseF mobility (Fig. 3b), which was in line with the fact that the biosynthesis and bioactivity of PQS are known to be mediated by the *las* and *rhl* systems[38]. In contrast, culture supernatants from mutants lacking the siderophore synthesis pathways still facilitated the mobility of TseF in a way similar to those from wild-type bacteria (Fig. 3b). Because PQS is implicated in iron acquisition as an iron-chelating compound[39,40], the role of TseF in iron sequestration likely is achieved by its interactions with the PQS signalling molecule.

We further examined the interactions between TseF and PQS by incubating the protein with immobilized PQS or HQNO (4-hydroxy-2-heptylquinoline N-oxide). After removing unbound protein by extensive wash, we found that TseF (but not the unrelated protein PA0533) can be robustly retained by PQS. In contrast, neither TseF nor PA0533 interacts with the PQS cognate HQNO (Fig. 3c). Furthermore, when nitrocellulose membranes spotted with different amounts of PQS or HHQ (4-hydroxy-2-heptylquinoline) were incubated in a solution containing His6-TseF, the protein was retained by PQS in a dose-dependent manner (Fig. 3d). The disassociation constant ($K_d$) between PQS and TseF was 0.33 μM as measured by isothermal titration calorimetry (ITC). In line with the result shown in Fig. 3d, the affinity between HHQ and TseF ($K_d = 25.71$ μM) is 78-fold lower than that between PQS and TseF ($K_d = 0.33$ μM) (Supplementary Fig. 5). The addition of $Fe^{3+}$ led to an increase in the affinity between PQS and TseF, with a $K_d$ of 0.15 μM, suggestive of a tighter complex in the presence of iron. Under the same binding condition, no binding of GST to PQS or PQS-$Fe^{3+}$ was detected (Fig. 3e). Taken together, these results establish that TseF specifically interacts with the iron-chelating extracellular signalling molecule PQS.

These results predict that the assimilation of iron from PQS-$Fe^{3+}$ by the PAΔ3FeΔ*tseF* mutant requires TseF. We tested this hypothesis by introducing a plasmid that expresses TseF in an isopropyl-β-D-1-thiogalactopyranoside (IPTG)-regulated manner into strain PAΔ3FeΔ*tseF*. In the absence of the iron chelator EDDHA, all bacterial strains grew indistinguishably (Fig. 3f left panel). In medium containing 5 μg ml$^{-1}$ EDDHA, PQS-$Fe^{3+}$ failed to permit the strain to grow when the expression of *tseF* was not induced; the induction of *tseF* expression allowed PQS-$Fe^{3+}$ to supply the iron required for the bacterium to grow (Fig. 3f, middle and right panels). When PQS-$Fe^{3+}$ was used as the sole iron source, the growth of the PAΔ3FeΔ*tseF* strain can also be restored by exogenously provided TseF protein but not GST (Supplementary Fig. 6). Notably, bacterial growth rates under this condition were considerably low as it took 24 h to observe significant replication.

**FptA/OprF are required for TseF-mediated iron acquisition.** The high-affinity binding among TseF, PQS and $Fe^{3+}$ suggest that these molecules form a complex. How is iron sequestrated in this complex delivered into the cell? We hypothesized that TseF engages receptor proteins on the bacterial surface to deliver the metal ion. To identify such putative receptors, we performed affinity chromatography with GST-TseF-coated beads against total cell lysates of *P. aeruginosa*. Proteins retained by GST-TseF were detected by silver staining after SDS–polyacrylamide gel electrophoresis (SDS–PAGE). Several proteins that appeared to specifically be retained by GST-TseF but not GST were obtained. The use of lysates from mutants lacking H3-T6SS or TseF itself did not change the proteins retained by GST-TseF (Fig. 4a). Mass spectrometric analysis identified the band of approximate 75 kDa

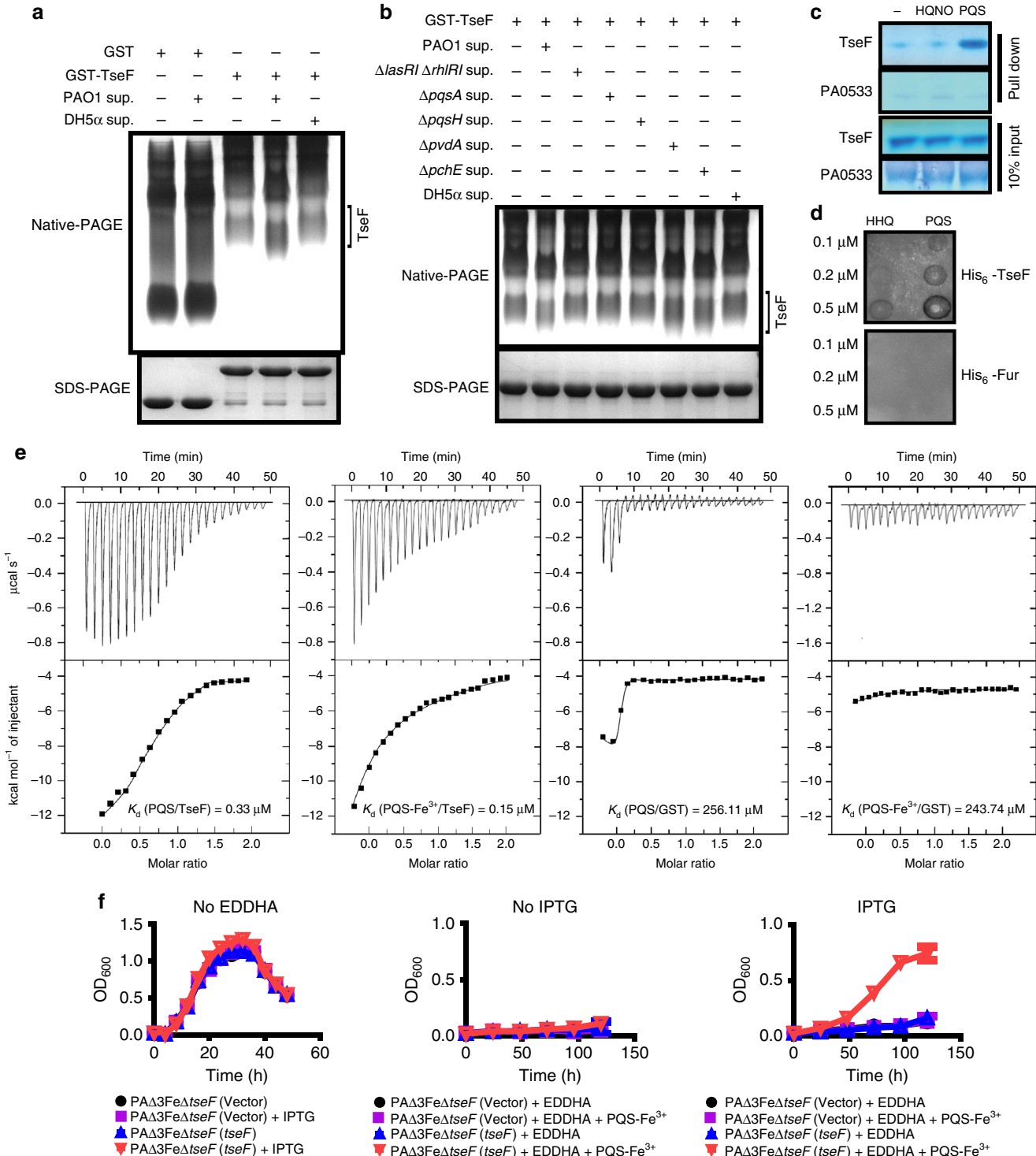

**Figure 3 | TseF interacts with PQS.** (**a**) Culture supernatant of *P. aeruginosa* causes an increase in the mobility of TseF. GST or GST-TseF was incubated with culture supernatant of *P. aeruginosa* or *E. coli* prior to native-PAGE or SDS–PAGE analysis. (**b**) The quinolone signal is required for causing TseF mobility shift. Culture supernatant of a series of *P. aeruginosa* mutants was tested for the ability to cause TseF mobility shift. Note that deletion of genes involved in the PQS signal production abolished the activity. (**c–e**) PQS binds TseF with high affinity. TseF but not PA0533 was retained by immobilized PQS (**c**); His$_6$-TseF but not His$_6$-Fur was retained by PQS spotted on membranes; under this condition, a higher amount of HHQ was required to retain similar amount of His$_6$-TseF (**d**); PQS binds to TseF with a $K_d$ of 0.33 μM, which was reduced to 0.15 μM when PQS-Fe was used (**e**). Neither PQS nor PQS-Fe$^{3+}$ interacted with GST. (**f**) Iron supplied by PQS requires TseF. The growth of the indicated bacterial strains was assessed in iron-deficient succinate MM (EDDHA: 5 μg ml$^{-1}$) without (left panel) or with 20 μM PQS-Fe$^{3+}$ (right panel). Note that only the expression of *tseF* rescued the growth of PAΔ3FeΔ*tseF* in the presence of PQS-Fe$^{3+}$ after >2 days of incubation. The curves represent three biological replicates; error bars are s.d. Full blots are shown in Supplementary Fig. 14.

as FptA, the Fe(III)-pyochelin receptor[41]; the 60 kDa protein was AtpA, the subunit A of the ATP synthase; the band around 40 kDa was the porin OprF, which is important for virulence and had been suggested to be involved in iron uptake[42,43]; and a band about 20 kDa specifically retained by GST-TseF was identified as PA4426, a protein of unknown function (Fig. 4a).

Next we purified the recombinant form of each of these proteins to validate the binding target for TseF. GST-TseF robustly interacted with $His_6$-OprF but not $His_6$-AtpA (Fig. 4b). It also reproducibly interacted with $FptA_{1–186}$, a truncation mutant of FptA in which the transmembrane domain was deleted to facilitate purification (Fig. 4c). It is worth noting that the interaction of TseF with FptA did not detectably undermine this receptor's binding to the $PCH-Fe^{3+}$ complex as examined by the $PCH-Fe^{3+}$-induced protein precipitation assay[44] (Supplementary Fig. 7). Thus TseF directly engages the pyochelin receptor FptA and the porin OprF.

To determine the role of these receptors in TseF-mediated iron uptake, we created strain $PA\Delta3Fe\Delta tseF\Delta fptA\Delta oprF$. This mutant grew normally in rich medium (Supplementary Fig. 8). However, this strain could not utilize $PQS-Fe^{3+}$ as the sole iron source for growth; introduction of a plasmid expressing FptA or OprF alone did not allow it to use iron supplied by $PQS-Fe^{3+}$ nor did the expression of TseF (Fig. 4d). Only when TseF and one of the receptors were simultaneously expressed did the bacteria gain the ability to utilize $PQS-Fe^{3+}$ to sustain growth (Fig. 4d). Taken together, these results establish that the role of TseF in iron acquisition is to bridge the iron-binding PQS to the receptors FptA or/and OprF.

**TseF is incorporated into OMVs for iron acquisition.** PQS is a highly hydrophobic molecule that is packaged into OMVs[45], which are involved in many signalling events[46,47]. In *Mycobacterium tuberculosis*, MVs have been reported to be involved in iron acquisition[48]. The *P. aeruginosa* OMVs contain most of the extracellular PQS molecules; these complexes have also been suggested to function in iron assimilation via a yet uncharacterized mechanism[49]. These observations prompted us to investigate whether TseF recruits PQS-containing OMVs for iron acquisition. We first examined the possibility that TseF is part of the OMV complex. Indeed, TseF-VSVG was present in OMVs purified from $\Delta tseF$ but not $\Delta tseF\Delta pqsH$ expressing the fusion protein (Fig. 5a and Supplementary Fig. 9), indicating that the association of TseF with OMVs occurs in a PQS-dependent manner. Consistent with earlier findings[50], flagellin FliC-VSVG is

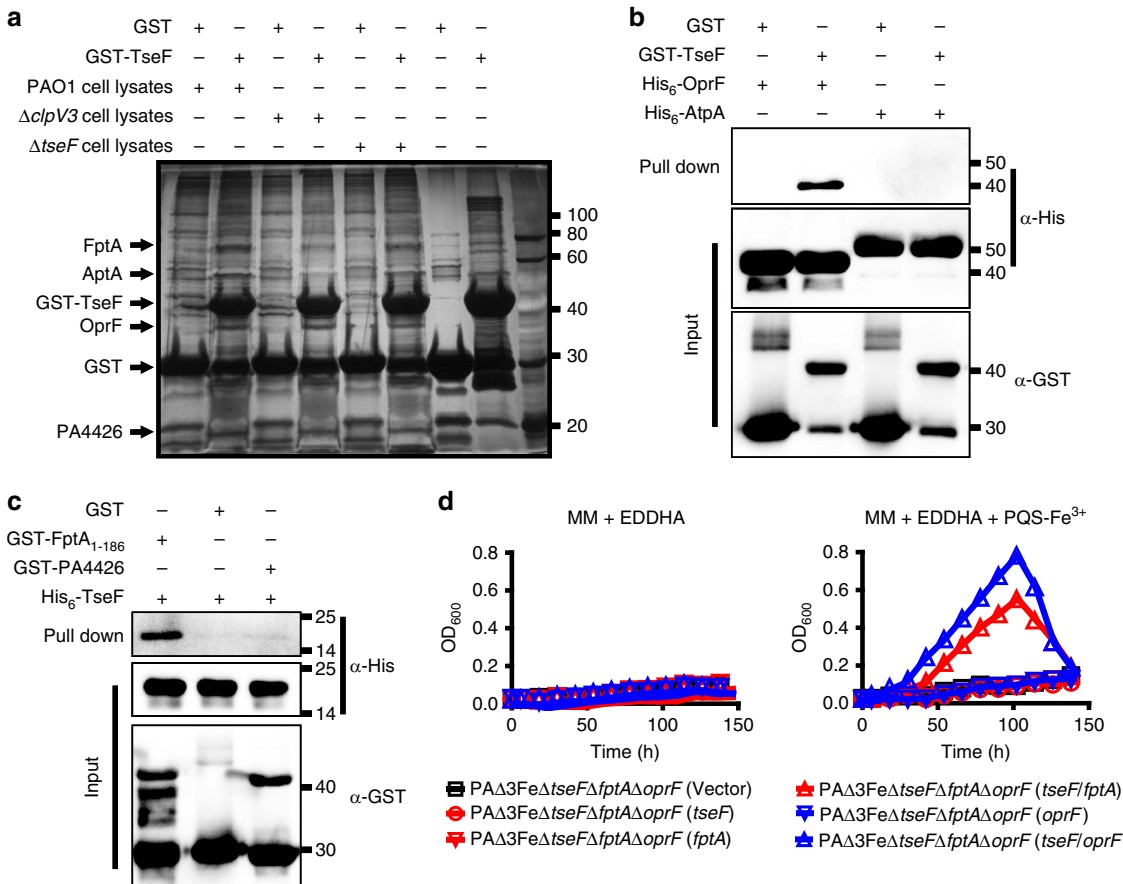

**Figure 4 | Porin OprF and the siderophore receptor FptA are essential for TseF-mediated iron uptake.** (**a**) The identification of OprF and FptA as the binding partners of TseF. Total cell lysates of the indicated bacterial strains were incubated with beads coated with GST or GST-TseF. After removing unbound proteins by extensively washing, proteins retained were resolved by SDS–PAGE followed by silver staining. Proteins specifically retained by GST-TseF were identified by mass spectrometry. (**b,c**) TseF directly interacts with OprF and FptA. GST or GST-TseF was incubated with $His_6$-tagged OprF or AtpA and the potential protein complex was captured by glutathione beads (**b**). Note that only $His_6$-OprF was retained by GST-TseF. A similar procedure was used to demonstrate direct binding between TseF and $FptA_{1–186}$ but not PA4426 (**c**). Full blots are shown in Supplementary Fig. 14. (**d**) The role of OprF and FptA in the uptake of iron supplied by $PQS-Fe^{3+}$. The growth of the indicated bacterial strains was assessed in succinate MM with EDDHA (5 µg ml$^{-1}$) without (left panel) or with (20 µM) $PQS-Fe^{3+}$ (right panel). Note that only in the presence of both TseF and one of the receptors were the bacteria able to use $PQS-Fe^{3+}$ to support its growth. The curves represent three biological replicates; error bars are s.d.

associated with OMVs prepared from both the $\Delta tseF$ and the $\Delta tseF\Delta pqsH$ strains. In contrast, the VgrG1b protein involved in substrate translocation was not detectably associated with purified vesicles (Fig. 5a and Supplementary Fig. 9). These results suggest that OMVs purified from wild-type *P. aeruginosa* would allow strain PAΔ3FeΔ*tseF* to grow under iron-deficient condition. Indeed, OMVs from wild-type but not the Δ*tseF* mutant promoted the growth of strain PAΔ3FeΔ*tseF* in the presence of $5\,\mu g\,ml^{-1}$ EDDHA (Fig. 5b). In agreement with this observation, OMVs from the Δ*tseF* strain cannot supplement iron requirement by this strain but gained such capacity when the gene was reintroduced (Fig. 5b).

The above results suggest that TseF and FptA/OprF are in the same iron acquisition pathway, which predicts that OMVs purified from culture supernatant of wild-type bacteria cannot supplement iron requirement by a mutant of PAΔ3FeΔ*tseF*

lacking both *fptA* and *oprF*. In agreement with this prediction, OMVs from strain PAO1 did not allow this mutant to grow in iron-deficient media, which was restored by expressing the PvdA system or one of the two TseF-binding proteins (Fig. 5c, left panel). As expected, although deletion of *tseF* had little effects on OMV production (Supplementary Fig. 10), OMVs from the Δ*tseF* mutant cannot provide iron required by the PAΔ3FeΔ*tseF*Δ*fptA*-Δ*oprF* mutant, even when FptA or OprF was expressed, but OMVs from a strain complemented with *tseF* regained the ability to supplement the iron needed for growth by these strains (Fig. 5c, middle and right panels). These results indicate that secreted TseF is incorporated into OMVs, where it interacts with iron-binding PQS to bring the metal ion to its receptors on the cell surface.

We further evaluated the role of TseF-mediated iron acquisition in the biology of *P. aeruginosa* by examining the

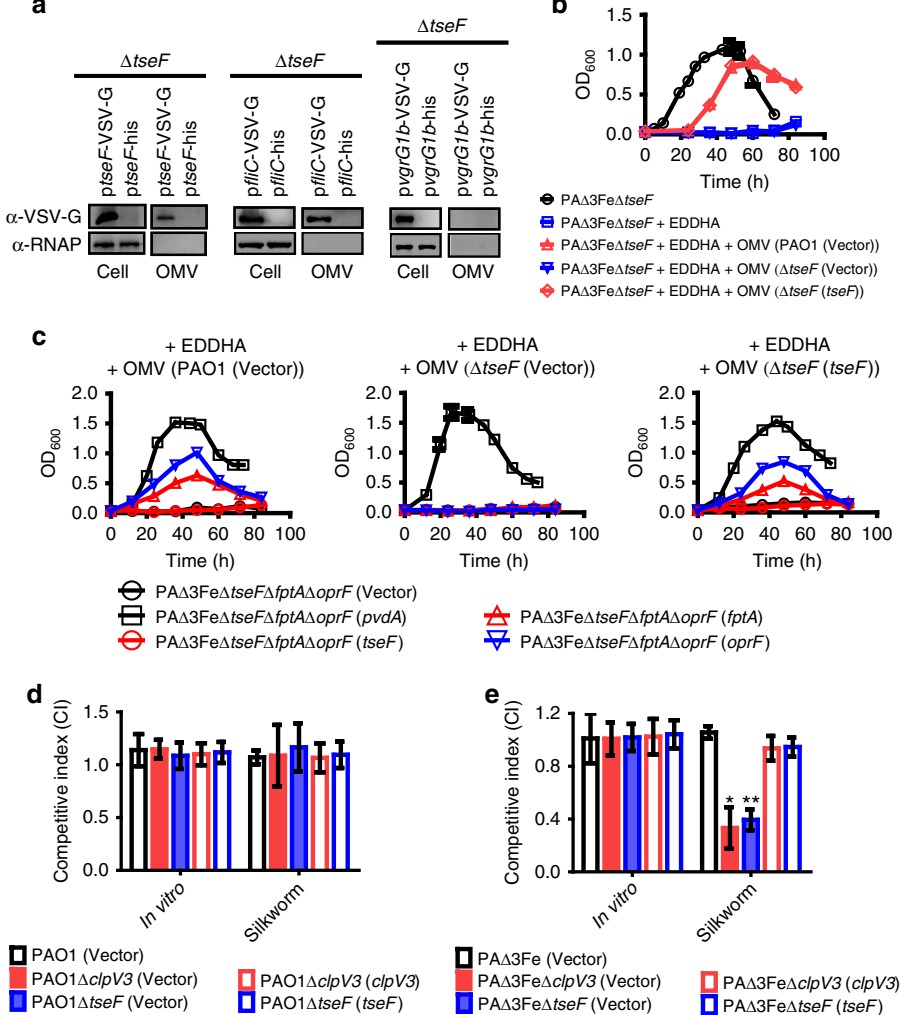

**Figure 5 | OMV complex is involved in TseF-mediated iron acquisition.** (**a**) TseF is incorporated into the OMV complex. OMVs prepared from the Δ*tseF* mutant expressing TseF-VSV-G, FliC-VSV-G or VgrG1b-VSV-G and the proteins of interest were probed. The cytosolic RNA polymerase was detected as a control. Full blots are shown in Supplementary Fig. 14. (**b**) TseF is required for supplementing the bacterium the iron by OMVs. Mutant PAΔ3FeΔ*tseF* grown in the presence of EDDHA ($5\,\mu g\,ml^{-1}$) was supplemented with OMVs ($0.01\,OD\,ml^{-1}$) prepared from bacterial strain with the indicated genotypes and bacterial growth was monitored by measuring the absorbance at $OD_{600}$. (**c**) Iron acquisition via OMVs requires OprF or FptA. OMVs from wild-type, Δ*tseF* or its complemented strain were used to supply iron for derivatives of mutant PAΔ3FeΔ*tseF*Δ*fptA*Δ*oprF*. Note that only those expressing *pvdA* or one of the binding partners (FptA and OprF) for TseF can use this iron source. (**d,e**) The TseF iron acquisition system is required for bacterial survival in a host. Equal amounts of the indicated bacterial strains and wild-type bacteria were co-inoculated onto silkworms and the survival of the bacteria was determined. Note that the strains lacking all the four iron acquisition systems are defective in competing against the wild-type bacteria in the host, which can be fully complemented by expressing the corresponding genes. Data shown were the average of three independent experiments and error bars indicate s.d. Student's *t*-test: *$P < 0.05$; **$P < 0.01$.

interactions of relevant mutants with silkworm, a host that can be used to examine the virulence of this pathogen[51]. Mutants defective in H3-T6SS or *tseF* were able to effectively compete with wild-type bacteria in media or in the host (Fig. 5d). Deletion of either *clpV3* (a core component of H3-T6SS) or *tseF* from mutant PAΔ3Fe resulted in mutants that cannot effectively compete against the wild type when co-inoculated in the host; complementation with the corresponding genes restored the wild-type phenotype (Fig. 5e). Indeed, as the major secretion apparatus of TseF, the H3-T6SS was also involved in utilization of PQS-Fe$^{3+}$, as the growth of the PAΔ3FeΔ*clpV3* mutant was reduced when PQS-Fe$^{3+}$ was used as the sole iron source, and such growth defect was restored by expressing *clpV3* (Supplementary Fig. 11). Thus iron uptake mediated by H3-T6SS is important for *P. aeruginosa* virulence in this host.

## Discussion

To cope with nutritional challenges, particularly those from their hosts, pathogenic bacteria have evolved many effective means to scavenge metal iron from the environment. Here we reveal a new mechanism for iron acquisition in *P. aeruginosa* involving type VI secretion, a QS signal, OMVs and a siderophore receptor.

The protein that integrates these distinct systems is TseF, which seems to be a substrate of H3-T6SS. Differing from YezP, a T6SS substrate involved in zinc uptake in *Y. pseudotuberculosis* that directly binds the metal ion[28], TseF appears to function by interacting with two classes of molecules: the PQS and the siderophore receptor FptA (Figs 3 and 4). At least three lines of evidence indicate that TseF, PQS and FptA are functioning in the same pathway. First, TseF directly interacts with FptA; second, PQS directly binds TseF; finally, iron supplied as a Fe$^{3+}$-PQS complex cannot be utilized by strains of *P. aeruginosa* lacking TseF, FptA or PQS (Figs 3f and 4d). The role of H3-T6SS and its substrate TseF in iron acquisition is further supported by the fact that the activity of H3-T6SS promoters (P$_{H3\text{-}T6SS\text{-}right}$, P$_{H3\text{-}T6SS\text{-}left}$ and P$_{vgrG3}$) in the PAΔ3Fe strain is induced by iron-deficiency conditions (Supplementary Fig. 12a,b). Furthermore, consistent with the notion that genes involved in iron acquisition are usually repressed by the ferric uptake regulator (Fur) for intracellular iron homeostasis[52], Fur negatively regulates the activity of H3-T6SS promoters (P$_{H3\text{-}T6SS\text{-}right}$, P$_{H3\text{-}T6SS\text{-}left}$ and P$_{vgrG3}$) and *tseF* expression (Supplementary Fig. 12c,d). Interestingly, besides being co-regulated with H3-T6SS by QS, the transcription of H2-T6SS is also repressed by Fur and iron[30]. Regulation of T6SS by Fur or iron has also been reported in *E. coli*, *Edwardsiella tarda*, *Burkholderia mallei* and *Burkholderia pseudomalleis*[53–55]. Finally, a T6SS in *Pseudomonas taiwanensis* was recently shown to be involved in the secretion of the iron chelator pyoverdine by a yet unidentified mechanism[56]. Together, these observations suggest that iron uptake may be a common function of T6SSs.

Our findings shed light into the role played by several molecules in iron metabolism of *P. aeruginosa*. PQS has a high affinity for iron and is able to induce the expression of genes for pyochelin biosynthesis[39]. Yet, its role in iron metabolism has been enigmatic. Our results reveal that iron associated with PQS is utilized by the bacterium in a TseF-mediated process. Similarly, OMVs have been implicated in iron acquisition but the underlying mechanism is not understood[49]. The highly hydrophobic PQS contributes to the generation of OMVs by intercalating into the outer membrane and inducing membrane curvature and is one of its integral components[45,57]. In *M. tuberculosis*, MVs were reported to directly participate in iron acquisition via the iron-binding compound mycobactin present in the vesicles[48]. Our observation that OMVs prepared from wild-type but not the Δ*tseF* mutant enabled the strain

defective in all four iron uptake systems to grow in the presence of EDDHA established that iron sequestered by these vesicles can be scavenged by the bacterium, validating the earlier speculation on the involvement of OMVs in iron assimilation by *P. aeruginosa*[49]. PQS carried by OMVs serves to traffic this signalling molecule within a population[45]. Given its ability to allow dispersion of PQS in an aqueous environment, it is possible that OMVs may enable long distance transport of this essential element, potentially allowing trans-feeding of bacteria localized at different sites of the host during infection. Finally, it has been shown that a porin (OprF) mutant was defective in the uptake of the siderophore–iron complex[43] but the underlying mechanism was unclear. OprF is a general outer membrane porin, which allows nonspecific diffusion of ionic species and small polar nutrients; it is also required for *P. aeruginosa* virulence through modulating the QS networks mediated by homoserine lactone and PQS[42]. We propose that OprF may function as an outer membrane conduit or receptor for TseF to facilitate the uptake of iron in complex with PQS.

Our results establish that iron acquisition can be achieved by the integration of molecules with distinct activities into a multi-component complex. Essential to this model is the T6SS substrate TseF that incorporates into OMVs whereby it interacts with the iron-binding PQS molecule (Fig. 6). Whereas the incorporation of TseF into OMVs is dependent on its interactions with PQS (Fig. 5a and Supplementary Fig. 9), the incorporation of PQS into OMVs does not require TseF (Supplementary Fig. 13). TseF also engages the receptors FptA and/or OprF to allow the cell to transfer the metal ion into the cytosol by an unknown mechanism (Fig. 6). Clearly iron acquisition by the PQS-TseF complex differs from those mediated by hydrophilic iron ligands. Other bacteria may employ similar mechanisms for iron acquisition. For example, in *M. tuberculosis* MVs containing the hydrophobic mycobactin siderophore can serve as an iron donor to support the replication of iron-starved mycobacteria[48]. Interestingly, the

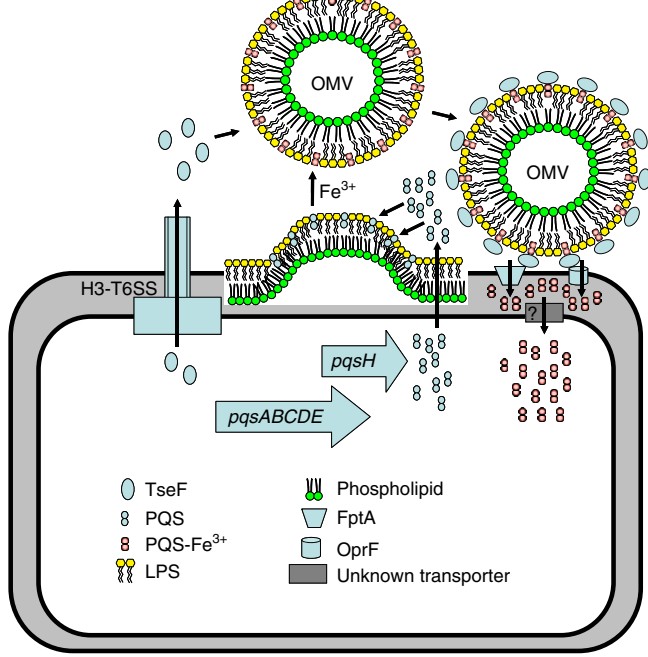

**Figure 6 | A model for TseF-mediated iron acquisition via the OMV complex.** We propose that TseF is exported by H3-T6SS and then incorporated into OMVs containing PQS-Fe$^{3+}$. Recognition of TseF by the cell surface receptors FptA or OprF would facilitate the transport of iron into the cell by a yet unidentified mechanism.

Esx-3 type 7 secretion system also participates in the assimilation of mycobactin-bound iron by secreting a pair of proteins belonging to the PE-PPE family (PE5–PPE4)[58,59]. It will be interesting to investigate whether the PE-PPE proteins are involved in the recruitment of MVs for iron acquisition by directly interacting with mycobactin. The presence of TseF homologues in a diverse array of bacteria suggests that this mechanism of iron acquisition may be widely employed.

## Methods

**Bacterial strains and growth conditions.** Bacterial strains and plasmids used in this study are listed in Supplementary Table 1. *E. coli* strains were grown at 37 °C in either Luria-Bertani (LB) broth or agar. *P. aeruginosa* strains were grown at 37 °C in either LB, tryptic soy broth (TSB), succinate MM[60] or M9 MM (Na$_2$HPO$_4$ 6 g l$^{-1}$, KH$_2$PO$_4$ 3 g l$^{-1}$, NaCl 0.5 g l$^{-1}$, NH$_4$Cl 1 g l$^{-1}$, MgSO$_4$.7H$_2$O 0.25 g l$^{-1}$, glucose 2 g l$^{-1}$). The *P. aeruginosa* PAO1 strain was the parent of all derivatives used in this study. To construct in-frame deletion mutants, the pK18*mobsacB* derivatives were transformed into relevant *P. aeruginosa* strains through *E. coli* S17-1-mediated conjugation and screened as previously described[61]. For overexpression or complementation in various *P. aeruginosa* strains, the pME6032 derivatives were transformed into relevant *P. aeruginosa* strains and induced by addition of 1 mM IPTG. Antibiotics were used at the following concentrations for *E. coli*: kanamycin, 50 μg ml$^{-1}$; tetracycline, 15 μg ml$^{-1}$; gentamicin, 15 μg ml$^{-1}$; and for *P. aeruginosa*: kanamycin, 50 μg ml$^{-1}$; chloramphenicol, 30 μg ml$^{-1}$; gentamicin, 100 μg ml$^{-1}$; tetracycline, 200 μg ml$^{-1}$ for plates or 160 μg ml$^{-1}$ for liquid growth.

**Transposon mutagenesis.** Random mini-Tn5 transposon mutagenesis was used to generate mutants in *P. aeruginosa* PAΔ3Fe as previously described with minor modifications[62]. The mini-Tn5 transposon was introduced into *P. aeruginosa* PAΔ3Fe by conjugation with the donor strain *E. coli* S17-1 λ-pir(pUTmini-Tn5-luxCDABE-Tc). *P. aeruginosa*-recipient cells were grown at 42 °C prior to biparental mating. Mixtures of the donor and recipient (1:2) were incubated on LB agar plates for 6–8 h at 37 °C. To select for *P. aeruginosa* transconjugants, cells were resuspended in PBS and plated onto LB agar plates containing 200 μg ml$^{-1}$ of tetracycline and 50 μg ml$^{-1}$ of kanamycin. The colonies with a transposon inserted should have resistance to tetracycline (to maintain the mini-Tn5 plasmid) and kanamycin (indicative of a recipient strain). Mutants were individually inoculated in succinate MM containing EDDHA (0.5 μg ml$^{-1}$) and the growth was monitored by measuring the absorbance at 600 nm. Mutants that failed to grow were retained for further analysis. To determine the insertion sites for mutants, genomic DNAs were SphI digested and circularized, followed by PCR using two outward-facing, transposon-specific primers listed in Supplementary Table 2. The PCR products were purified, sequenced and mapped to the *P. aeruginosa* genome.

**Plasmid construction.** Primers used in this study are listed in Supplementary Table 2. To construct the knock-out plasmid for deletion, the *tseF* (*PA2374*) gene, the 757-bp upstream fragment and the 769-bp downstream fragment flanking *tseF* were amplified with primer pairs TseF up F/TseF up R and TseF low F/TseF low R. The upstream and downstream PCR fragments were ligated by overlap PCR, and the resulting PCR products were inserted into the XbaI/HindIII sites of the suicide vector pK18*mobsacB*. The gentamicin resistance cassette amplified from p34s-Gm was subsequently inserted into the same HindIII site to yield the knock-out plasmid pK-*tseF*. The knock-out plasmids pK-*vgrG3* (*PA2373*), pK-B3C3 (*PA2365-2366*), pK-*clpV3* (*PA2371*), pK-*pvdA* (*PA2386*), pK-*pchE* (*PA4226*), pK-*feoB* (*PA4358*), pK-*oprF* (*PA1777*), pK-*fptA* (*PA4221*), pK-*pqsA* (*PA0096*), pK-*pqsH* (*PA2587*) and pK-*fur* (*PA4764*) were constructed in similar manners by using primers listed in Supplementary Table 2.

To construct the complementation plasmid pME6032-*tseF*, PCR-amplified *tseF* was cloned into the EcoRI and XhoI sites of plasmid pME6032, giving rise to the plasmid pME6032-*tseF*. To construct the complementation plasmids pME6032-*tseF*-*fptA* and pME6032-*tseF*-*oprF*, primers TseF F/TseF SacI R, FptA F/FptA XhoI R and OprF F/OprF XhoI R were used to amplify *tseF*, DNA fragments encompassing the Shine-Dalgarno sequence and the structural part of *fptA* and *oprF* from genomic DNA of PAO1, respectively. The PCR products were inserted into pME6032, using the EcoRI/SacI, SacI/XhoI and SacI/XhoI sites, respectively. To construct the complementation plasmid pBBR1MCS-5-*clpV3*, primer ClpV3 EcoRI/ClpV3 R low was used to amplify *clpV3* fragment from genomic DNA of PAO1. The PCR product of *clpV3* was digested with EcoRI/BglII and inserted into the EcoRI/BamHI sites of pBBR1MCS-5 resulting in the plasmid pBBR1MCS-5-*clpV3*. All other complementation plasmids were constructed by the same procedure as the plasmid pME6032-*tseF* or pBBR1MCS-5-*clpV3*.

To construct pME6032-*tseF-VSVG*, primer TseF F/TseF-VSVG R was used to amplify the *tseF* gene and the PCR product was digested with EcoRI/BglII and inserted into similarly digested pME6032 to generate pME6032-*tseF-VSVG*. The plasmids pME6032-*hcp3-VSVG* (*PA2367*), pME6032-*vgrG3-VSVG*, pME6032-*vgrG1a-VSVG* (*PA0091*), pME6032-*vgrG1b-VSVG* (*PA0095*) and

pME6032-*fliC-VSVG* (*PA1092*) were constructed in similar manners by using primers listed in Supplementary Table 2. These recombinant plasmids were then transformed into relevant *P. aeruginosa* strains by electroporation.

Plasmids pET28a-*tseF*, pET28a-*fur*, pET28a-*oprF* and pET28a-*atpA* (*PA5556*) were constructed as follows. Primers Fur F NdeI/Fur R HindIII, TseFE F/TseFE R, OprF BamHI F/OprF XhoI R and AtpA BamHI F/AtpA XhoI R were used to amplify *tseF*, *fur*, *oprF* and *atpA* gene fragments from PAO1 genome, respectively. The PCR products of *fur*, *tseF*, *oprF* and *atpA* were digested with NdeI/HindIII, NdeI/HindIII, BamHI/XhoI and BamHI/XhoI and inserted into the same sites of pET28a to generate pET28a-*fur*, pET28a-*tseF*, pET28a-*oprF* and pET28a-*atpA*, respectively. To express GST-tagged proteins, primers TseF BamHI F/TseF XhoI R, PA4426 BamHI F/PA4426 XhoI R, FptA BamHI F/FptA 186 XhoI R and PA0533 BamHI/PA0533 XhoI were used to amplify the *tseF*, *PA4426*, *fptA$_{1-186}$* and *PA0533* genes from genomic DNA of PAO1. The PCR products of relevant genes were digested with BamHI/XhoI and inserted into the BamHI/XhoI sites of pGEX6P-1 resulting in plasmids pGEX6p-1-*tseF*, pGEX6p-1-*PA4426*, pGEX6p-1-*fptA$_{1-186}$* and pGEX6p-1-*PA0533*. The integrity of the insert in all constructs was confirmed by DNA sequencing.

**Chromosomal fusion reporters and β-galactosidase assays.** The *H3-T6SS left-lacZ*, *H3-T6SS right-lacZ* and *vgrG3-lacZ* transcriptional fusions were constructed by PCR amplification of the 508, 508 and 1307 bp upstream DNA region from the *lip3*, *hsiB3* and *vgrG3* genes by using primer pairs P2364-300F/P2364-300R, P2365-300F/P2365-300R and P2373-1322/P2373 low, respectively (Supplementary Table 2). PCR amplification products from each of the upstream regions were cloned into the pMini-CTX::*lacZ* vector[63], yielding a range of *lacZ* reporter constructs as listed in Supplementary Table 1. Promoter fragments were integrated at the CTX phage attachment site (*attB*) in wild-type strain PAO1 and the relevant mutant strains. The unmarked transcriptional fusion strains were then constructed by Flp-catalysed excision of the Tc$^r$ marker following the established protocols[63]. For β-galactosidase assays, overnight bacterial cultures were diluted 1:500 in TSB. Growth and β-galactosidase activity were monitored by harvesting samples at different time points. β-Galactosidase activity was measured according to the Miller method based on *o*-nitrophenyl-β-D-galactopyranoside hydrolysis and was expressed in Miller units[64].

**Purification of recombinant proteins and western blotting.** GST- and His$_6$-tagged recombinant proteins were purified as described[28]. For western blottings, samples resolved by SDS–PAGE were transferred onto polyvinylidene difluoride membranes. After blocking with 5% (w/v) BSA in PBS buffer containing 0.2% Tween 20, membranes were incubated with the appropriate primary antibody: anti-VSVG (Santa Cruz Biotechnology, USA), 1:5,000; anti-RNA pol β (Santa Cruz Biotechnology), 1:200; anti-His (Santa Cruz Biotechnology), 1:200; and anti-GST (Zhongshan Golden Bridge Biotechnology, Beijing, China), 1:2,000. The membrane was washed three times in TBST buffer (50 mM Tris, 150 mM NaCl, 0.05% Tween 20, pH 7.4) and incubated with 1:10,000 dilution of horseradish peroxidase-conjugated secondary antibodies (Shanghai Genomics) for 1 h. Signals were detected using the ECL plus kit following the manufacturer's specified protocol.

**Protein secretion assay.** Secretory proteins were isolated using methods described previously[65] with minor modifications. In brief, relevant *P. aeruginosa* strains expressing TseF-VSV-G were allowed to grow in TSB medium at 37 °C under agitation based on their growth status, until OD$_{600}$ reached 2.5. The cells were removed, and supernatant was filtered through a 0.22 μm-pore-size filter and an Amicon Ultra-15 Centrifugal fliter (100 KD, Millipore). For each 100 ml of the culture supernatant fraction, 15 ml of 100% trichloroacetic acid and 3 ml of 1% sodium deoxycholate were added, and the fractions were incubated on ice overnight before centrifugation at 21,800 *g* and 4 °C for 60 min. The resulting protein pellet was washed with 90% ice-cold acetone three times. The protein pellets were resuspended in SDS protein sample buffer and incubated at 100 °C for 10 min. For western blotting analysis, 1 ml culture was centrifuged and the total proteins were prepared by dissolving the cell pellet with 100 μl SDS sample buffer. All samples were normalized to the OD$_{600}$ of the culture and volume used in preparation. Secretion assays for Hcp3 was carried out by a similar procedure.

**GST pull-down assay.** The GST pull-down assay was performed as previously described with minor modifications[66]. For screening TseF interaction partners in *P. aeruginosa*, stationary phase *P. aeruginosa* cells grown in M9 MM (250 ml) were collected and lysed by sonication in lysis buffer (10 ml PBS supplemented with 5 mM MgCl$_2$, 150 U DNase I, 2 mM PMSF, 2 mM DTT and 1% Triton X-100). Cleared cell lysates were incubated with 40 μg purified GST-TseF on a rotator at 4 °C overnight, and 40 μl prewashed glutathione-Sepharose beads (Novagen) were added to the reactions. After another 4 h of incubation at 4 °C, the beads were washed five times with TEN buffer (100 mM Tris-Cl (pH 8.0), 10 mM EDTA, 300 mM NaCl). Proteins associated with beads were solubilized with SDS sample buffer, separated by SDS–PAGE and detected by silver staining (Bio-Rad). Gel slices containing individual protein bands were excised, digested with trypsin and analysed by matrix-assisted laser desorption/ionization/mass spectrometry.

To verify the interactions between GST-TseF and VSVG-tagged VgrGs expressed in *P. aeruginosa*, 10 μg GST-TseF was mixed with 2–5 mg VgrG-expressing cell lysates on a rotator for 2 h at 4 °C and 40 μl prewashed glutathione beads were added to the reactions. After another 2 h of incubation at 4 °C, the beads were washed six times with TEN buffer containing 500 mM NaCl. Retained proteins were detected by immunoblot with anti-VSVG antibody after SDS–PAGE.

To verify protein–protein interactions with purified proteins, 10 μg purified GST-fusion protein was mixed with 10 μg His-tagged proteins in PBS for 2 h at 4 °C. After adding 30 μl of prewashed glutathione beads, binding was allowed to proceed for 2 h. The beads were then washed five times with TEN buffer containing 500 mM NaCl. Retained proteins were detected by immunoblotting after SDS–PAGE.

**Detection of TseF-binding proteins in cell-free supernatants.** Stationary phase *P. aeruginosa* and *E. coli* supernatants grown in M9 MM were filtered through 0.45 μm-pore-size filters (Pall Life Sciences, USA) and incubated with GST-TseF or GST-coated glutathione beads (Novagen) at 4 °C. M9 medium was treated in similar manner to serve as a negative control. After 2 h incubation, the beads were washed with PBS three times and GST fusion proteins were eluted from the glutathione beads with elution buffer according to the manufacturer's instructions. Eluted proteins were resolved by SDS–PAGE or Native-PAGE and visualized by Coomassie brilliant blue staining to compare the mobility shift between supernatant-treated and untreated proteins.

**LC-MS analysis of compounds bound to TseF.** Bacterial cells were allowed to grow for ∼24 h in TSB medium based on their growth status, until OD$_{600}$ reached 1.9–2.0. The supernatant of the bacterial culture (5 ml) was extracted with the same volume of ethyl acetate. The ethyl acetate extracts were dried using a speedvac concentrator and re-dissolved in 50 μl of methanol before these were used in next steps. The binding of TseF (100 μg) to this extract was performed in 500 μl of 10 mM Hepes, pH 7.4, 100 mM NaCl and 10 μl of chemical extracts at 4 °C for 6 h. As a control, the TseF protein was replaced by GST in the binding assay. The binding system was then filtered through a 3K centrifugal filter device (Amicon Ultra-15) to remove the protein. The filtered liquid was dried using the speedvac concentrator before 50 μl of methanol added. The dissolved part containing the chemicals was analysed by UPLC (Waters ACQUITY, USA) coupled with the MicroTOF-MS system (Bruker Daltonics GmbH, Bremen, German). Reverse-phase chromatography was performed in 2.1 × 150 mm$^2$ columns (Waters, BEH C18, 1.7 μm, USA) at a flow rate of 250 μl min$^{-1}$. Samples dissolved in acetonitrile were eluted by gradient mobile phase (5% to 95% acetonitrile with 0.1% formic acid in water) over 30 min. The raw LC-MS data were then analysed by using the software provided by Bruker Corporation.

**Fat western blotting.** The binding capacity of TseF to different 2-alkyl-4(1*H*)-quinolone (AQ) molecules were evaluated by fat western blotting as reported[67]. Briefly, AQs were solubilized in chloroform as a stock solution of 10 mM, and a minimum of 10 μl containing 0.1, 0,2 or 0.5 μM of AQs were spotted onto nitrocellulose (NitroBind, MSI, USA). The dotted membrane was dried at 24 °C for 1 h and the nitrocellulose was then incubated with 3% (w/v) fatty acid-free BSA (A-6003, isolated by cold ethanol precipitation; Sigma) in TBST (10 mM Tris (pH 8.0), 140 mM NaCl and 0.1% (v/v) Tween 20) for 1 h and then placed in a solution containing the His-tagged proteins (TseF or Fur) diluted in TBST (0.5 mg ml$^{-1}$) at 4 °C overnight with shaking. The membrane was washed with TBST three times for 10 min each and then incubated with anti-His monoclonal antibody (Santa Cruz Biotechnology, USA) diluted at 1:200 in TBST for 1 h at 24 °C. The incubated membrane was then washed three times for 10 min in TBST at 24 °C and then incubated with 1:10,000 dilution of horseradish peroxidase-conjugated secondary antibodies (Shanghai Genomics) for 1 h. Signals were detected using the ECL Plus Kit (GE Healthcare, Piscataway, NJ) following the manufacturer's specified protocol.

**Binding of proteins to the PQS affinity probe.** PQS and HQNO contain primary alkyl groups that could be coupled to an Affi-gel 10 (Bio-Rad, Hercules, CA, USA) matrix[68]. The compound–matrix coupling was performed by incubation of 1.5 ml Affi-gel media with 0.3 mg of each compound and incubated at 4 °C for 2 h following the Bio-Rad's manual. The gel matrix loaded with compounds was then incubated with the purified TseF protein (final concentration 0.1 mg ml$^{-1}$) at 4 °C for another 2 h to detect the interactions. After stringent washing in high ionic-strength solutions (500 mM NaCl, 1% Triton X-100 in Tris-buffered saline), proteins associated with the gel matrix were solubilized with SDS sample buffer, resolved by SDS–PAGE and visualized by Coomassie staining. The purified PA0533 protein was used as a negative control.

**Isothermal titration calorimetry.** ITC experiments for quinolone and proteins were performed following previously described procedures[69]. Briefly, the experiments were carried out using a VP-ITC MicroCalorimeter (MicroCal, Northampton, MA, USA) in the ITC buffer (20 mM Tris, pH 7.4, 150 mM NaCl, 10% glycerol (v/v), 5% DMSO (v/v)) at 25 °C. Concentrations of compound solutions in DMSO were determined by the weight. Final compound concentrations were achieved by diluting 1:20 (v/v) in the ITC buffer. Protein concentration was measured based on the absorbance at 280 nm using a NanoDrop 2000 UV-Vis Spectrophotometer. Protein samples were prepared by dialysis in the ITC buffer. Quinolone (0.05 mM) was filled into the syringe compartment (volume 100 μl) while the protein solution (5 μM) was dispensed into the microcalorimetric cell (volume 850 μl). After temperature equilibration, 3 μl of quinolone was titrated in every 2.2 min into the TseF-containing cell under constant stirring. All compounds were also titrated into ITC buffer alone, and the resulting heat of dilution was subtracted from the experimental curves. As a negative control, quinolone was titrated into the GST protein dispersion. In binding assays between iron ions and proteins, apo-proteins were prepared by dialysis against 250 μM EDTA and 5 mM *o*-phenanthroline in 50 mM HEPES (pH 8.0) before dialysis in the ITC buffer. The iron-binding proteins Fur and transferrin (Sigma) were used as positive controls, while the GST protein was used as a negative control. For the binding assay between $Fe^{2+}$ and Fur, 0.05 mM ferrous chloride was injected into 5 μM protein. For the binding assay between $Fe^{3+}$ and apo-transferrin, 150 mM NaCl in the ITC buffer was replaced by 130 mM NaCl and 20 mM NaHCO$_3$.

**Isolation and quantification of OMVs.** OMVs were isolated and quantified as described[45] with minor modifications. *P. aeruginosa* strains were grown overnight in TSB and diluted 1:100 in fresh TSB broth containing 100 μM FeCl$_3$ until OD$_{600}$ reached 2.5 at 37 °C under agitation. After the bacteria were pelleted at 6,000*g* for 20 min, the supernatant fraction was filtered through a 0.45-μm vacuum filter, and the filtrate was concentrated with ultrafiltration (100 kDa centrifugal filter device (Millipore)). The retentate was filtered again through a 0.22-μm vacuum filter to remove any remaining bacteria. The resulting filtrate was subjected to ultracentrifugation at 265,000*g* for 60 min in a Beckman Coulter 70.1 rotor. For quantitative purposes, OMV preparations were subjected to two buffer (50 mM HEPES, pH 8.0) exchanges with 100-kDa centrifugal filter devices (Millipore). Protein content of the OMVs, which has been previously used to standardize OMV preparations, was used as a measure of OMV levels by evaluating the absorbance at 220 nm.

**Protein precipitation assay.** The $PCH-Fe^{3+}$-mediated protein precipitation assay was performed as previously described[44]. In brief, the $PCH-Fe^{3+}$ complexes would induce a conformational change in the FptA$_{1-186}$ protein and finally result in protein precipitation. In all, 0.02 mM purified FptA$_{1-186}$ protein was added to $PCH-Fe^{3+}$ solutions (0.1 and 1 mM) with or without 0.02 mM purified TseF protein. The mixed protein and compound were incubated at room temperature for 30 min before centrifugation at 15,000 r.p.m. for 10 min. Then the supernatants were loaded onto 12% SDS–PAGE gels for protein analysis. The transcriptional regulatory protein PA0533 was used as a negative control.

**RNA extraction and quantitative real-time PCR (qRT–PCR).** RNA extraction was performed following the manufacturer's instructions of the AllPrep DNA/RNA Mini Kit (Qiagen, Hilden, Germany), which facilitated the extraction of DNA and RNA from the same cells. Briefly, cultured *P. aeruginosa* cells were pelleted by centrifugation at 4,000*g* for 10 min before lysis with lysozyme and proteinase K. The DNA was filtered out by the DNA column provided in the kit and DNase treatment. The RNA was reverse-transcribed into cDNA using SuperScript III reverse transcriptase (Invitrogen, Carlsbad, CA) and random primers. Three replicates were extracted for the qRT–PCR. The PCR was conducted using the Kapa SYBR Fast qPCR Kit (Kapa Biosystems, Woburn, MA) for cDNA with a Mx3000P qPCR machine (Agilent Technologies, Palo Alto, CA). The 20-μl qPCR reaction contained 10 μl of 2 × Master Mix, 0.5 μl of template and 1 pmol μl$^{-1}$ of forward and reverse primers. The cycling parameters were 5 min at 95 °C and then 40 cycles of 15 s at 95 °C, 15 s at 50 °C and 15 s at 60 °C. The relative expression levels of the *tseF* and *vgrG3* genes in different strains were normalized to that of the housekeeping gene *rpoD* (*PA0576*).

**Insect infection experiments.** A *lacZ* reporter gene was transferred to the neutral phage attachment site (*attB*) of the *P. aeruginosa* chromosome as follows: 513 nucleotides of the plasmid pME6032 *tac* promoter were cloned into the pMini-CTX::*lacZ* vector, and the resulting plasmid was then transferred to the *P. aeruginosa* chromosome (*attB* site) by mating and selection for tetracycline resistance. The selectable marker was removed by transient expression of the Flp recombinase from plasmid pFLP2, which was then cured by counter-selection on sucrose plates[63]. The resulting *P. aeruginosa* strains PAO1 *attB*::Ptac-*lacZ* and PAΔ3Fe *attB*::Ptac-*lacZ* were confirmed to grow as well as wild-type PAO1 and PAΔ3Fe in both *in vitro* and *in vivo* competition assays.

Infection in silkworms was performed as described previously with minor modifications[51]. Briefly, exponential phase *P. aeruginosa* strains grown in TSB were washed three times and re-suspended with PBS. In each experiment, a *P. aeruginosa* strain contained *lacZ* reporter inserted at the neutral phage attachment site (produces blue colonies on X-Gal plates; 60 μg ml$^{-1}$) was mixed 1:1 with a *P. aeruginosa* strain without *lacZ* (producing white colonies on X-Gal plates). For infections, 50 μl of a mixed bacteria suspension of $1 \times 10^7$ colony-forming unit (CFU) of *P. aeruginosa* per ml was injected the haemocoel of day 3,

fifth instar larvae of *Bombyx mori* (*Nistari* strain). At 24 h postinfection, five larvae of each group were anaesthetized on ice, and the haemolymph was collected in an ice-cold eppendorf tube containing 1.5 µl propylthiouracil. The haemolmph samples were subsequently diluted serially 10-fold with sterilized PBS and titered for CFU on LB plates containing X-gal and scored both for total CFU and ratio of blue-to-white bacteria. The competitive indices was calculated as the ratio of white-to-blue colonies in the output sample divided by the ratio of white-to-blue colonies in the input sample. The animals were selected randomly for each test group.

**Statistical analysis.** All experiments were performed in triplicate and repeated on two different occasions. Data are expressed as mean ± s.d. Differences between frequencies were assessed by Student's *t*-test (bilateral and unpaired) using a *P* value of <0.05 as statistically significant. Shapiro–Wilk test and Levene's test were performed using the SPSSv13.0 software (SPSS, Chicago, IL), respectively, to examine the normality of the data and the homogeneity of variances.

**Data availability.** The authors declare that all the relevant data supporting the findings of this study are available within the article and its Supplementary Information files or from the corresponding author on request.

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

## Acknowledgements

We thank Dr Herbert P. Schweizer at Colorado State University for kindly providing the plasmid pMini-CTX::*lacZ*. We also thank Dr Zhiqiang Lu and Mr Li Ma at Northwest A&F University for technical assistance in insect infection. This work was supported by the National Natural Science Foundation of China 31670053 and 31170121 (to X.S.), 31370150 (to Y.W.), 31200040 (to J.C.), Fundamental Research Funds for the Central Universities, Northwest A&F University 2452015100 (to X.S.), China Postdoctoral Science Foundation 2014M562455 (to J.C.) and a grant from China Ocean Mineral Resources Research and Development Association COMR-RDA13SC01 (to P.-Y.Q.).

## Author contributions

J.L., Y.W. and X.S. designed research; J.L., W.Z., J.C., X.Y., K.Z. and Y.W. performed research; G.W. and P.-Y.Q. contributed new reagents/analytic tools; J.L., W.Z., Y.W., Z.-Q.L. and X.S. analysed data; J.L., W.Z., Z.-Q.L. and X.S. wrote the paper.

## Additional information

**Competing interests:** The authors declare no competing financial interests.

