## [Peer Review File · Nature Communications]

Reviewers' comments:

Reviewer #1 (Remarks to the Author):

The manuscript findings are: A protein secreted by a type VI secretion system is needed for iron uptake independent of siderophores and reduced Fe transport systems in *Pseudomonas*. This protein, TseF does not seem to bind Fe directly but it interacts with PQS which itself chelates Fe. PQS-Fe can supply Fe to Fe deficient cells and this requires TseF and the membrane proteins OprF or FptA. Furthermore, TseF directly interacts with OprF and with FptA, a siderophore receptor. At least one of these two membrane proteins is required for soluble PQS-TseF Fe mediated delivery. TseF is incorporated into OMVs and OMVs can deliver Fe in a TseF, PfrF/fptA dependent manner. This pathway of Fe delivery is relevant during infection.

The results are well presented and the manuscript is well written although there are few grammatical errors.

Specific Comments

According to the model TseF is secreted and then interacts with PQS-Fe in OMVs and the membrane receptors to facilitate Fe delivery. However, although the interaction of TseF and soluble PQS is well supported there is no evidence that incorporation of TseF into OMV is dependent on the interaction with PQS as suggested by the model. The experiment showed in Fig5a could be done with OMV produced by a PQS synthesis mutant to address this question. Conversely, is PQS incorporation into OMV dependent on TseF? Do OMV from the TseF mutant contain PQS? Perhaps TseF allows PQS incorporation into OMV were it can access distant Fe. Would interaction of TseF with FptA undermine this receptor's interaction with siderophore-Fe complexes?

Please add in the methods section the methods used for Fe- protein interactions experiments shown in Supp Fig 3. Were these experiments done with Fe+3 or Fe+2 . A positive control for Fe binding should be included in this experiment before concluding that TseF does not bind Fe.

Minor

L23 remove allows it to

L32 a heme oxygenase

L54 importing iron

L108 as expected

L176 with the iron-chelating

L224 the *M. tuberculosis* membrane vesicles originate from the plasma membrane so technically they are not OMVs is better to refer to these as membrane vesicles.

L248 complemented instead of complementary

L256 were/was

L176 interacts with the...

L318-319 this line is confusing: there is no evidence that esx3 secretion is connected with membrane vesicles although both esx3 and MVs are connected with iron acquisition.

G. Marcela Rodriguez

Reviewer #2 (Remarks to the Author):

The manuscript NCOMMS-16-21636 entitled "A *Pseudomonas* T6SS effector recruits PQS-containing outer membrane vesicles for iron acquisition" led to characterize a novel iron acquisition

pathway coupling the H3-T6SS effector TseF, PQS, outer membrane vesicles (OMVs), and the outer membrane channels FptA and OprF.

Since iron acquisition is particularly challenging, bacteria have evolved many ways to scavenge iron from the environment. *P. aeruginosa* competes for iron by producing the high affinity siderophores pyoverdine and pyochelin, as well as hemophores, and they can also import xenosiderophores released by other bacteria. The authors provide herein strengthened data leading to the characterization of an additional mechanism of iron acquisition via OMVs, PQS and the essential H3-T6SS effector TseF that was shown to interact directly with the pyochelin receptor FptA and the major outer membrane porin OprF.

The manuscript is well written, the figures including several controls are convincing and clear, and the corresponding analyses and conclusions are sound. The authors add a graphical abstract (Figure 6), summarizing the main data of their study as a model, which will benefit readers outside the discipline.

Experiments were nicely conducted and data were presented in a logical way. This study is based on strong and multiple data, originating from many various assays (transposon mutagenesis, molecular cloning, plasmids and chromosomal fusion reporters construction, production of recombinant proteins, protein secretion, GST pull-down, fat western blotting, binding assays and LC-MS identification, ITCs, isolation and quantification of OMVs, infection assays...). The data that are presented very clearly, led to the discovery and characterization of unexpected functions of a new H3-T6SS effector; which was shown to bind OMVs, PQS-Fe(III) and the outer membrane components FptA and OprF.

Some points would strengthen the paper further:

1/ *P. aeruginosa* T6SS systems are poorly introduced. Previous literature demonstrating relationships between QS, H3-T6SS and/or iron should be mentioned, at least at the expression level. This should also be discussed.

2/ Discussion section: the authors claimed that the functions of TseF fills in the several gaps in understanding of the role of several molecules in iron metabolism. While this claim appears justified for PQS, the function of OprF in this mechanism remains elusive. What is the function of OprF in this mechanism? OprF is a water channel with a rather cationic selectivity. In addition to its function as porin, OprF was also shown to act as a structural proteibn and an environmental sensor. How can OprF function by engaging TseF to import iron in complex with PQS? May OprF be a receptor of TseF? This should be discussed with regards to OprF functions, particularly in line with previous work showing relationships between OprF and QS (Las, Rhl and PQS).

3/ Fig. 1C: an excess of PvdA seems to be detrimental to *P. aeruginosa* PA Δ 3Fe Δ tseF (pvdA) after 20 h of growth in MM + 0.5 μ g/ml EDDHA. This should be discussed.

4/ Supplementary Fig. 4: the first peak, identified as PQS strongly decreases in samples that were incubated with YseF. What about the next peak that seems to be slightly increased in this condition? In the same vein, HHQ seems also to retain His6-TseF, in lower concentrations as PQS does. Binding assays between HHQ and TseF would be informative. Sentence (L. 170-171) should be accordingly modified, and this possibility should be discussed.

5/ pa2374 and vgrG3 are predicted as an operon, but this is only predicted. Effect of Fur should be assayed on tseF directly, not only on vgrG3.

Some typos: L. 54 and L. 90: *aeruginosa* and not *aeruginosa*

Reviewer #3 (Remarks to the Author):

In this very interesting paper, the authors describe how a *Pseudomonas aeruginosa* mutant in which the three iron uptake systems pyoverdine, pyochelin and the Fe(II) Feo system has been deleted is still able to grow in minimal medium, but, as expected not in medium supplemented with EDDHA. Using a transposon mutagenesis approach, they obtained a mutant in the PA2374

gene, which is in the close vicinity of the type VI (H3T6SS) gene cluster. They also demonstrate that this small protein is a substrate of H3T6SS (although not exclusively). The PA2374 encoded protein is also demonstrated to bind PQS with high affinity. Further, the authors show that PA2374-PQS-Fe³⁺ binds to two outer membrane proteins, OprF, the major porin, and the pyochelin receptor, FptA (at least with the N-terminal 1-186 fragment). Finally, the protein-PQS complex is delivered to OMVs where it can deliver iron to cells.

The manuscript reads very well and the data are extremely interesting. I read it with great interest and I think that it will be a milestone in the field of iron uptake by *P. aeruginosa*. I have therefore only some minor remarks:

- the authors should be aware that a third "siderophore" mediated iron uptake system has been described in *P. aeruginosa* recently by Gi et al. (2015) in Scientific Reports. This low affinity iron chelator is nicotianamine and it is probably involved in the transport of zinc as well.

- I am a bit surprised by the interaction between the residues 1-186 of FptA with PA2374. This N-terminal domain represents the plug domain of the TonB-dependent receptor, which is buried within the receptor pore.

Reviewers' comments:

Reviewer #1 (Remarks to the Author):

The manuscript findings are: A protein secreted by a type VI secretion system is needed for iron uptake independent of siderophores and reduced Fe transport systems in *Pseudomonas*. This protein, TseF does not seem to bind Fe directly but it interacts with PQS which itself chelates Fe. PQS-Fe can supply Fe to Fe deficient cells and this requires TseF and the membrane proteins OprF or FptA. Furthermore, TseF directly interacts with OprF and with FptA, a siderophore receptor. At least one of these two membrane proteins is required for soluble PQS-TseF Fe mediated delivery. TseF is incorporated into OMVs and OMVs can deliver Fe in a TseF, PfrF/fptA dependent manner. This pathway of Fe delivery is relevant during infection.

The results are well presented and the manuscript is well written although there are few grammatical errors.

Specific Comments

According to the model TseF is secreted and then interacts with PQS-Fe in OMVs and the membrane receptors to facilitate Fe delivery. However, although the interaction of TseF and soluble PQS is well supported there is no evidence that incorporation of TseF into OMV is dependent on the interaction with PQS as suggested by the model. The experiment showed in Fig5a could be done with OMV produced by a PQS synthesis mutant to address this question. Conversely, is PQS incorporation into OMV dependent on TseF? Do OMV from the TseF mutant contain PQS? Perhaps TseF allows PQS incorporation into OMV were it can access distant Fe.

Response: We thank the reviewer for this very insightful point. As suggested, we have detected the incorporation of TseF into OMVs produced by a PQS synthesis mutant ($\Delta pqSH$) (**Supplementary Figure 9**). The results show that the incorporation of TseF into OMV is dependent on the interaction with PQS, as we didn't detect TseF in OMVs prepared from the $\Delta pqSH$ mutant (**Supplementary Figure 9**).

We have also determined the presence of PQS in OMVs prepared from the $\Delta tseF$ mutant using HPLC-MS. The results show that the OMVs prepared from the $\Delta tseF$ mutant contain as much PQS as those from the wild-type bacteria (**Supplementary Figure 13**), suggesting that the incorporation of PQS into OMV occurs independent of TseF.

All these points have been included in the text (Lines 251-253 and Lines 346-348).

Would interaction of TseF with FptA undermine this receptor's interaction with siderophore-Fe complexes?

Response: To answer this question, we assessed the effect of TseF on the interaction between FptA and PCH-Fe³⁺ complexes with a protein precipitation assay reported in an earlier study (Xiao et al., 2006). We found that the PCH-Fe³⁺ complexes induce a

conformational change in the FptA₁₋₁₈₆ protein, as suggested by the precipitation assay: 0.02 mM FptA₁₋₁₈₆ protein was added to PCH-Fe³⁺ solutions (0.1 and 1 mM) with or without 0.02 mM TseF; the mixture was incubated at room temperature for 30 min followed by centrifugation; the supernatants were subjected to SDS-PAGE. The results demonstrate that TseF does not detectably affect the precipitation of FptA induced by the PCH-Fe³⁺ complexes (**Supplementary Figure 7**).

Please note that the PCH-Fe³⁺ complexes did not induce the precipitation of PA0533, a transcriptional regulatory protein served as a negative control (**Supplementary Figure 7**). This result has been included in the text (Lines 228-231) and the method has been described (Lines 598-605).

Relevant references:

Xiao et al., 2006. MvfR, a key *Pseudomonas aeruginosa* pathogenicity LTTR-class regulatory protein, has dual ligands. **Mol. Microbiol.** 2006, 62(6), 1689–1699.

Please add in the methods section the methods used for Fe- protein interactions experiments shown in Supp Fig 3. Were these experiments done with Fe⁺³ or Fe⁺². A positive control for Fe binding should be included in this experiment before concluding that TseF does not bind Fe.

Response: The method used for Fe-protein interactions experiments shown in **Supplementary Figure 3** has been added in the methods section (Lines 576-583). These experiments were done with Fe⁺³ and Fe⁺² and have been indicated in the text. A positive control (Fur for Fe⁺² and human transferrin for Fe⁺³) for Fe binding has also been included in this experiment. The results showed that there was no significant binding between TseF and Fe³⁺/Fe²⁺. A revised **Supplementary Figure 3** with the results and controls has been provided.

Minor

L23 remove allows it to

Response: Corrected.

L32 a heme oxygenase

Response: Corrected.

L54 importing iron

Response: Corrected.

L108 as expected

Response: Corrected.

L176 with the iron-chelating

Response: Corrected.

L224 the *M. tuberculosis* membrane vesicles originate from the plasma membrane so technically they are not OMVs is better to refer to these as membrane vesicles.

Response: Corrected.

L248 complemented instead of complementary

Response: Corrected.

L256 were/was

Response: Corrected.

L176 interacts with the...

Response: Corrected.

L318-319 this line is confusing: there is no evidence that esx3 secretion is connected with membrane vesicles although both esx3 and MVs are connected with iron acquisition.

Response: We have revised these sentences as follow: “For example, in *M. tuberculosis* membrane vesicles containing the hydrophobic mycobactin siderophore can serve as an iron donor to support the replication of iron-starved mycobacteria⁴⁸. Interestingly, the Esx-3 type 7 secretion system also participates in the assimilation of mycobactin-bound iron by secreting a pair of proteins belonging to the PE-PPE family (PE5–PPE4)^{59,60}. It will be interesting to investigate whether the PE-PPE proteins are involved in the recruitment of MVs for iron acquisition by directly interacting with mycobactin. (Lines 352-358)”.

Reviewer #2 (Remarks to the Author):

The manuscript NCOMMS-16-21636 entitled “A *Pseudomonas* T6SS effector recruits PQS-containing outer membrane vesicles for iron acquisition” led to characterize a novel iron acquisition pathway coupling the H3-T6SS effector TseF, PQS, outer membrane vesicles (OMVs), and the outer membrane channels FptA and OprF.

Since iron acquisition is particularly challenging, bacteria have evolved many ways to scavenge iron from the environment. *P. aeruginosa* competes for iron by producing the high affinity siderophores pyoverdine and pyochelin, as well as hemophores, and they can also import xenosiderophores released by other bacteria. The authors provide herein strengthened data leading to the characterization of an additional mechanism of iron acquisition via OMVs, PQS and the essential H3-T6SS effector TseF that was shown to interact directly with the pyochelin receptor FptA and the major outer membrane porin OprF.

The manuscript is well written, the figures including several controls are convincing and clear, and the corresponding analyses and conclusions are sound. The authors add a

graphical abstract (Figure 6), summarizing the main data of their study as a model, which will benefit readers outside the discipline.

Experiments were nicely conducted and data were presented in a logical way. This study is based on strong and multiple data, originating from many various assays (transposon mutagenesis, molecular cloning, plasmids and chromosomal fusion reporters construction, production of recombinant proteins, protein secretion, GST pull-down, fat western blotting, binding assays and LC-MS identification, ITCs, isolation and quantification of OMVs, infection assays...). The data that are presented very clearly, led to the discovery and characterization of unexpected functions of a new H3-T6SS effector; which was shown to bind OMVs, PQS-Fe(III) and the outer membrane components FptA and OprF.

Some points would strengthen the paper further:

1/ *P. aeruginosa* T6SS systems are poorly introduced. Previous literature demonstrating relationships between QS, H3-T6SS and/or iron should be mentioned, at least at the expression level. This should also be discussed.

Response: We have added these points in the instruction and discussion sections as suggested. The revised text reads: “H1-T6SS is known to target prokaryotic cells by delivering multiple bacteriolytic toxins into target cells, providing a competitive advantage to *P. aeruginosa* in polymicrobial communities²⁹. In contrast, H2- and H3-T6SS target both prokaryotic and eukaryotic cells by using the PldA and PldB trans-kingdom effectors^{23,24}. Moreover, both H2- and H3-T6SS contribute to the virulence of *P. aeruginosa* in infection models of animal and plant^{30,31}. The expression of *P. aeruginosa* T6SSs is differentially regulated by quorum sensing (QS). Whereas the expression of H1-T6SS is suppressed by both the homoserine lactone transcription factor LasR and the 4-hydroxy-2-alkylquinoline (HAQ) transcriptional regulator MvfR, the expression of H2- and H3-T6SS is positively regulated by MvfR and LasR³¹. In addition, PqsE, a key component of the MvfR regulon, is required for the expression of part of H3-T6SS but not H2-T6SS³¹. Despite these progresses in their regulation, the function of H2- and H3-T6SS still remains largely unknown. (Lines 75-87)”

“Interestingly, besides co-regulated with H3-T6SS by QS, the transcription of H2-T6SS is also repressed by Fur and iron similar to H3-T6SS⁵³. Regulation of T6SS by Fur or iron has also been reported in *E. coli*, *Edwardsiella tarda*, *Burkholderia mallei* and *Burkholderia pseudomalleis*⁵⁴⁻⁵⁶. Finally, a T6SS in *Pseudomonas taiwanensis* was recently shown to be involved in the secretion of the iron chelator pyoverdine by a yet unidentified mechanism⁵⁷. Together, these observations suggest that iron uptake should be a general function of T6SS. (Lines 311-317)”.

Relevant references:

Sana, T.G. *et al.* The second type VI secretion system of *Pseudomonas aeruginosa* strain PAO1 is regulated by quorum sensing and Fur and modulates internalization in epithelial cells. *J. Biol. Chem.* **287**, 27095-27105 (2012)

Lesic B, Starkey M, He J, Hazan R, Rahme LG. Quorum sensing differentially regulates *Pseudomonas aeruginosa* type VI secretion locus I and homologous loci II and III, which are required for pathogenesis. **Microbiology**. **155**, 2845-55. (2009)

Brunet, Y.R., Bernard, C.S., Gavioli, M., Llobès, R. & Cascales, E. An epigenetic switch involving overlapping *fur* and DNA methylation optimizes expression of a typeVI secretion gene cluster. **PLoS Genet**. **7**, e1002205 (2011).

Chakraborty, S., Sivaraman, J., Leung, K.Y. & Mok, Y.K. Two-component PhoB-PhoR regulatory system and ferric uptake regulator sense phosphate and iron to control virulence genes in type III and VI secretion systems of *Edwardsiella tarda*. **J. Biol. Chem.** **286**, 39417-39430 (2011).

Burtnick, M. N. & Brett, P.J. *Burkholderia mallei* and *Burkholderia pseudomallei* cluster 1 type VI secretion system gene expression is negatively regulated by iron and zinc. **PLoS One** **8**, e76767. (2013).

Chen WJ, Kuo TY, Hsieh FC, Chen PY, Wang CS, Shih YL, Lai YM, Liu JR, Yang YL, Shih MC. Involvement of type VI secretion system in secretion of iron chelator pyoverdine in *Pseudomonas taiwanensis*. **Sci Rep**. **6**, 32950. (2016)

2/ Discussion section: the authors claimed that the functions of TseF fills in the several gaps in understanding of the role of several molecules in iron metabolism. While this claim appears justified for PQS, the function of OprF in this mechanism remains elusive. What is the function of OprF in this mechanism? OprF is a water channel with a rather cationic selectivity. In addition to its function as porin, OprF was also shown to act as a structural protein and an environmental sensor. How can OprF function by engaging TseF to import iron in complex with PQS? May OprF be a receptor of TseF? This should be discussed with regards to OprF functions, particularly in line with previous work showing relationships between OprF and QS (Las, Rhl and PQS).

Response: We thank the reviewer for this important point. In fact, the involvement of OprF in iron transport has been reported in *P. aeruginosa* and *M. smegmatis* (Meyer, 1992; Jones and Niederweis, 2010). It has been speculated that bacteria use high-affinity uptake systems for acquisition of iron under iron-limited conditions, and use low-affinity uptake systems such as porins for acquisition of iron under high-iron conditions. In *P. aeruginosa*, the porin OprF is the uptake route for iron complexes of desferriferrioxamine B, desferriferrioxamine E, desferriferriochrysin and desferriferrirococin, siderophores produced by other bacteria or fungi (Meyer, 1992). In *M. smegmatis*, the Msp porins are involved in the acquisition of soluble iron under high-iron conditions possible by acting as the outer membrane conduit of low-affinity iron acquisition systems (Jones and Niederweis, 2010). Indeed, the Msp porins are also the main mediators of copper uptake across the outer membrane in *M. smegmatis* and *M. tuberculosis* (Speer et al., 2013).

According to the reviewer's suggestion, we have further discussed this point as follows: "Finally, a porin-OprF mutant was defective in the uptake of the siderophore-iron complex⁴³ but the underlying mechanism has been long elusive. OprF is a general outer membrane porin, which allows nonspecific diffusion of ionic species and small polar nutrients; it is also required for *P. aeruginosa* virulence through modulating the quorum-sensing networks mediated by homoserine lactone and PQS⁴². We propose

that OprF functions as an outer membrane conduit or a receptor of TseF to facilitate the uptake of iron in complex with PQS. (Lines 336-342)”

Relevant references:

Meyer JM. Exogenous siderophore-mediated iron uptake in *Pseudomonas aeruginosa*: possible involvement of porin OprF in iron translocation. *J Gen Microbiol.* 1992 May;138(5):951-8.

Jones CM, Niederweis M. Role of porins in iron uptake by *Mycobacterium smegmatis*. *J Bacteriol.* 2010, 192(24):6411-7.

Speer A, Rowland JL, Haeili M, Niederweis M, Wolschendorf F. Porins increase copper susceptibility of *Mycobacterium tuberculosis*. *J Bacteriol.* 2013, 195(22):5133-40.

Fito-Boncompagni L, Chapalain A, Bouffartigues E, Chaker H, Lesouhaitier O, Gicquel G, Bazire A, Madi A, Connil N, Véron W, Taupin L, Toussaint B, Cornelis P, Wei Q, Shioya K, Déziel E, Feuilloley MG, Orange N, Dufour A, Chevalier S. Full virulence of *Pseudomonas aeruginosa* requires OprF. *Infect Immun.* 2011, 79(3):1176-86.

3/ Fig. 1C: an excess of PvdA seems to be detrimental to *P. aeruginosa* PA Δ 3Fe Δ tseF (pvdA) after 20 h of growth in MM + 0.5 μ g/ml EDDHA. This should be discussed.

Response: Thank you for raising this important question. Actually this phenomenon has been documented by several different groups in *Pseudomonas* culture with the succinic acid medium (Chiadò et al., 2013; Mirleau et al., 2000; Sulochana et al., 2014). The decline of bacterial growth at the stationary phase was caused by the increase of pH, which had been discussed by Chiadò et al as follows: “In each test conducted with succinic acid, the OD₆₂₀ fell when the pH was higher than 8.4, while the culture with glucose remained stable during the whole stationary phase, with a recorded pH of about 6.5. In other papers, it has been reported that *P. fluorescens* cultures grown in succinic acid reach high pH values, of about 8.0–8.8, during the stationary phase (Appanna and St. Pierre, 1996). It is possible that the depletion of succinic acid, buffered at pH 7.0, left an excess of OH⁻ in the M78 medium, with a resulting increase in pH and a detrimental effect on the viability of the microorganism.” (Chiadò et al., 2013). This reference has been cited.

Relevant references:

Chiadò A, Varani L, Bosco F, Marmo L. Opening Study on the Development of a New Biosensor for Metal Toxicity Based on *Pseudomonas fluorescens* Pyoverdine. *Biosensors (Basel).* 2013, 3(4):385-99.

Mirleau P, Delorme S, Philippot L, Meyer J, Mazurier S, Lemanceau P. Fitness in soil and rhizosphere of *Pseudomonas fluorescens* C7R12 compared with a C7R12 mutant affected in pyoverdine synthesis and uptake. *FEMS Microbiol Ecol.* 2000, 34(1):35-44.

Sulochana MB, Jayachandra SY, Kumar SK, Dayanand A. Antifungal attributes of siderophore produced by the *Pseudomonas aeruginosa* JAS-25. *J Basic Microbiol.* 2014, 54(5):418-24.

Appanna V.D., St. Pierre M. Cellular response to a multiple-metal stress in *Pseudomonas fluorescens*. *J. Biotechnol.* 1996, 48:129–136.

4/Supplementary Fig. 4: the first peak, identified as PQS strongly decreases in samples that were incubated with TseF. What about the next peak that seems to be slightly increased in this condition? In the same vein, HHQ seems also to retain His₆-TseF, in lower concentrations as PQS does. Binding assays between HHQ and TseF would be informative. Sentence (L. 170-171) should be accordingly modified, and this possibility should be discussed.

Response: This is a very good suggestion. The peak next to PQS was identified to be 2-nonyl-4-hydroxyquinoline (NHQ; m/z=272.2). While the peak corresponding to PQS was strongly decreased in samples that had been incubated with TseF, the peak corresponding to NHQ seems to be slightly increased in this condition. Because HPLC only allows for semi-quantitative analysis, the slight increase of NHQ would not challenge our conclusion that TseF binds PQS. Clearly, NHQ will serve as the best negative control for the binding assay. However, this compound is not commercially available to us, so instead we used commercial available HQNO and HHQ as controls.

Since HHQ seems also to retain His₆-TseF (Fig. 3d), binding assays between HHQ and TseF were performed with ITC as suggested (**Supplementary Figure 5**). The results reveal a weak binding between HHQ and TseF (K_d=25.71 μM), which is significantly lower than that between PQS and TseF (K_d=0.33 μM). Sentence (L. 170-171) have been accordingly modified and discussed.

5/ pa2374 and vgrG3 are predicted as an operon, but this is only predicted. Effect of Fur should be assayed on tseF directly, not only on vgrG3.

Response: Effect of Fur on *tseF* as well as *vgrG3* expression has been assayed with qRT-PCR as suggested. The results show that similar to *vgrG3*, the expression of *tseF* in *fur* mutant is significantly higher than the wild-type PAO1 strain (**Supplementary Figure 12d**).

Some typos: L. 54 and L. 90: aeruginosa and not aeruginosa

Response: Corrected.

Reviewer #3 (Remarks to the Author):

In this very interesting paper, the authors describe how a *Pseudomonas aeruginosa* mutant in which the three iron uptake systems pyoverdine, pyochelin and the Fe(II) Feo system has been deleted is still able to grow in minimal medium, but, as expected not in medium supplemented with EDDHA. Using a transposon mutagenesis approach, they obtained a mutant in the PA2374 gene, which is in the close vicinity of the type VI (H3T6SS) gene cluster. They also demonstrate that this small protein is a substrate of H3T6SS (although not exclusively). The PA2374 encoded protein is also demonstrated to bind PQS with high affinity. Further, the authors show that PA2374-PQS-Fe³⁺ binds

to two outer membrane proteins, OprF, the major porin, and the pyochelin receptor, FptA (at least with the N-terminal 1-186 fragment). Finally, the protein-PQS complex is delivered to OMVs where it can deliver iron to cells.

The manuscript reads very well and the data are extremely interesting. I read it with great interest and I think that it will be a milestone in the field of iron uptake by *P. aeruginosa*. I have therefore only some minor remarks:

- the authors should be aware that a third "siderophore" mediated iron uptake system has been described in *P. aeruginosa* recently by Gi et al. (2015) in Scientific Reports. This low affinity iron chelator is nicotianamine and it is probably involved in the transport of zinc as well.

Response: Thank you for providing this important information. We have mentioned this issue in the discussion section as suggested as follows: "Recently, a nicotianamine siderophore mediated iron uptake system was identified to be essential for the growth of *P. aeruginosa* in airway mucus". (Lines 57-59)

Relevant references:

Gi M, Lee KM, Kim SC, Yoon JH, Yoon SS, Choi JY. A novel siderophore system is essential for the growth of *Pseudomonas aeruginosa* in airway mucus. *Sci Rep.* 2015, 5:14644.

- I am a bit surprised by the interaction between the residues 1-186 of FptA with PA2374. This N-terminal domain represents the plug domain of the TonB-dependent receptor, which is buried within the receptor pore.

Response: The involvement of the plug domain in the binding of an interacting protein is not unprecedented. For example, colicin E2 utilizes the BtuB outer membrane receptor for entering the cell. According to the solved structure of the colicin E2R135-BtuB Complex, 28 residues of E2R135 interact with 29 residues of BtuB, including six residues of the plug domain (Thr55, Gln56, Asn57, Leu63, Ser64, and Ser65) (Sharma et al., *JBC*, 2007).

Relevant references:

Sharma O, Yamashita E, Zhalnina MV, Zakharov SD, Datsenko KA, Wanner BL, Cramer WA. Structure of the complex of the colicin E2 R-domain and its BtuB receptor. The outer membranecolicin translocon. *J Biol Chem.* 2007, 282(32):23163-70.

REVIEWERS' COMMENTS:

Reviewer #1 (Remarks to the Author):

The manuscript is suitable for publication.

Please include Serafini A et al. 2013. PlosOne Vol 8 (10) e78351, the first publication that documented the role of M. tuberculosis Esx3 in Fe uptake and Zn homeostasis.

Reviewer #2 (Remarks to the Author):

The authors have satisfactorily addressed all the points raised in a previous round, strengthening further the original manuscript. To my opinion, this paper adds a lot of novelties in the field of iron acquisition pathway, which is of major interest.

REVIEWERS' COMMENTS:

Reviewer #1 (Remarks to the Author):

The manuscript is suitable for publication.

Please include Serafini A et al. 2013. Plos One Vol 8 (10) e78351, the first publication that documented the role of *M. tuberculosis* Esx3 in Fe uptake and Zn homeostasis.

Response: Thank you very much for your constructive comments, which has greatly improved our manuscript. Since the manuscript already contains several references about the role of Esx3 in Fe uptake and Zn homeostasis in *M. tuberculosis* (e.g. ref. 5, 59 and 60), it's not very necessary to add one more.

Reviewer #2 (Remarks to the Author):

The authors have satisfactorily addressed all the points raised in a previous round, strengthening further the original manuscript. To my opinion, this paper adds a lot of novelties in the field of iron acquisition pathway, which is of major interest.

Response: Thank you very much for your constructive comments, which has greatly improved our manuscript.